# SeedThink: Test-Time Control via Seed-Thought Initialization

## Abstract

Large reasoning models (LRMs) achieve impressive performance through extended chains of thought, but this substantially increases inference overhead, making efficiency a critical bottleneck. In this paper, we first show that initializing the reasoning process with high-quality seed thoughts can steer the model away from unproductive "overthinking" and produce more efficient reasoning trajectories. Critically, we find that the optimal granularity of this seed — from a high-level outline to a detailed solution — depends on problem difficulty. Motivated by this, we propose SeedThink, a novel framework that adaptively selects the seed granularity based on an estimate of problem difficulty. Specifically, SeedThink features two core innovations: (1) a **difficulty-aware seeding policy that dynamically generates seed thoughts** to reduce repetitive verification and prune unproductive branches; and (2) **seamless integration with enhanced speculative decoding**, where seed thoughts are repurposed as a model-free draft corpus to achieve dual-path acceleration — shorter reasoning traces and faster token generation. Our experiments show that SeedThink significantly reduces inference costs while largely preserving performance. Notably, our method achieves up to a 4.1× end-to-end speedup and a 68% reduction in generation length with minimal accuracy degradation, highlighting the promise of adaptive initialization for balancing reasoning quality and efficiency.

## 1 INTRODUCTION

Large language models increasingly benefit from test-time compute, a phenomenon known as test-time scaling (Snell et al., 2024; Muennighoff et al., 2025). Recent large reasoning models (LRMs), such as OpenAI o1 (Jaech et al., 2024), DeepSeek-R1 (Guo et al., 2025), and Qwen3 (Yang et al., 2025), demonstrate that extended chains of thought (Wei et al., 2022a), paired with reflection, backtracking, and self-verification techniques, significantly enhance performance on complex reasoning problems. Given a query, these models generate a deliberate think trajectory before producing a concise final solution, thereby yielding better results across mathematical and scientific reasoning benchmarks (Cobbe et al., 2021; Hendrycks et al., 2021; He et al., 2024; Rein et al., 2024; Lewkowycz et al., 2022).

However, these advances introduce two significant and interconnected challenges: the inherent **inefficiency** and **path dependence** of test-time computation. While extended chains of thought can substantially improve performance, they often lead to "overthinking" — expending extra tokens on repetitive verification, unproductive branching, and exploration long after a sufficient solution has been found (Chen et al., 2024; Cuadron et al., 2025). A single *Thinking* trajectory is typically five times longer than a direct *No-Thinking* response, even on simple problems where both modes produce the same correct answer (Zhang et al., 2025a; Fan et al., 2025; Liang et al., 2025).

A natural approach to improving efficiency is the mode-switching paradigm, widely studied in recent works (Zhang et al., 2025a;b; Li et al., 2025a; Luo et al., 2025; Liang et al., 2025; Ha et al., 2025; Xing et al.). It switches between the efficient *No-Thinking* mode for easy problems and the effective *Thinking* mode for hard ones. However, this paradigm has a critical limitation: its efficacy declines sharply as problem difficulty increases. It gains efficiency only on problems that are already within the *No-Thinking* mode's capability, failing to address the core inefficiency and path dependence of "Thinking" on truly challenging tasks. An emerging direction to bridge this gap is to integrate the

**Question: What is the smallest positive multiple of 450 whose digits are all zeroes and ones?**

**No Thinking**

<think>Think finished</think>
We are asked to find the smallest positive multiple of 450 whose... (90 tokens)

**Step 1**: Find prime factorization of 450. 450=5*90... (320 tokens)

**Step 2**: Search for the smallest number made of only 0s and1s, ending in 00, and divisible by 9. (650 tokens)

**Step 3**: Check the smallest number made of only 0s and 1s that is divisible by ... (800 tokens)

Final answer: boxed{100100}

Total Tokens: **1852** ✗

*Efficient but ineffective*

**Thinking**

<think>Okay, so I need to find the smallest multiple of 450 whose digits are ... (1870 tokens)
But wait ... (1700 tokens)
Let me check... (2400 tokens)
Let me think of another approach that... </think> (2600 tokens)

To determine the smallest positive multiple of 450... we must consider the divisibility rules and the digit composition constraints... (710 tokens)

Answer: boxed{11111111100}

Total Tokens: **9694** ✓

*Inefficient but effective*

**Thinking with Seed**

Here is a *seed solution* for this question...
**Step 1**: Use Prime Factor...
**Step 2**: Apply Digit Cons...
Total Tokens: **342**

<think>Okay, I should follow the detailed steps provided in the *seed solution*...
**Step 1**: Use Prime Factor...
**Step 2**: Apply Digit Constrains...
**Step 3**: Find final ans...</think>

Answer: boxed{11111111100}

Total Tokens: **5209** ✓

*Efficient and effective*

Figure 1: Qwen3-32B's Thinking and No-Thinking mode generating process for an example from the MATH (Hendrycks et al., 2021) benchmark. No Thinking executes a single-pass, myopic trajectory that under-explores the solution space, while Thinking broadens coverage via multi-pass, multi-angle checks, but spends many tokens on unproductive trajectories. Thinking with Seed initializes the reasoning process with a concise seed that steers the trajectory to productive regions, shortens the reasoning path, and prunes unnecessary branches, thereby improving the efficiency of model reasoning.

two modes or multiple agents into a cooperative pipeline (Fan et al., 2025; Pan et al., 2025; Wang et al., 2025d).

The inefficiency is further compounded by the highly path-dependent nature of the reasoning trajectory — the same thought budget can produce dramatically different outcomes depending on how the reasoning process is initialized. Both empirically and intuitively, early steps constrain what the model explores next, so the initial state of the reasoning process can determine whether the model converges quickly on a correct explanation or wanders into unproductive regions, and can even affect the model's instruct-following and safety performance (Fan et al., 2025; Lin et al., 2025; Wu et al., 2025c).

To address these challenges, in this paper, we investigate a **test-time steering technique via seed-thought initialization**: how to start the thinking process so that limited test-time compute is steered toward productive regions of the search space. We operationalize this idea using seed thoughts: concise, high-quality initial trajectories that shape the early steps of reasoning. As shown in Figure 1, *No-Thinking* is fast but myopic, while standard *Thinking* broadens coverage at a significant token cost. Our ***Thinking with Seed* initializes reasoning with a concise seed that captures essential pivot reasoning anchors** (e.g., subgoals, key lemmas, variable definitions, candidate equation forms), which can steer exploration toward productive regions and prune unnecessary branches.

The **thought generation process in LRMs can be conceptualized as a non-convex search through language space**. This search is initialized with the user prompt and a seed thought. The resulting thought trajectories — sequences of tokens evaluated by either process reward models (PRMs) or outcome reward models (ORMs) — represent the model's paths to a solution. Therefore, our method fundamentally alters the search dynamics: rather than starting from scratch, the *Thinking with Seed* mode is initialized at a point already oriented toward a promising basin in the search space. This head start significantly shortens the reasoning trajectory. Furthermore, the seed acts as a lightweight greedy backbone that effectively constrains the branching factor during subsequent exploration, yet remains revisable. By pruning vast regions of the search space that are unproductive from the outset, the method improves efficiency and mitigates path dependence by steering the reasoning process toward high-probability solution basins, thereby increasing the chances of rapid and correct convergence.

Having established the importance of seed thoughts, a natural question arises: *does a one-size-fits-all seed exist?*. In practice, seed granularity is a direct lever on the trade-off between guidance

and flexibility. In our pilot experiments, we consistently observed that for easier problems, more detailed seeds collapse the search and improve efficiency without increasing error, because even imperfect specifics rarely mislead the model. For harder problems, however, higher-level seeds are preferable, because they provide strategic scaffolding and flexibility without locking the model into potentially misleading, concrete steps. These findings motivate a **difficulty-aware seeding policy that adjusts seed granularity** to dynamically shape the reasoning trajectory, instead of relying on fixed initialization strategies.

The utility of the generated seed thoughts extends beyond guiding the reasoning search. We further show that the two-stage generation paradigm naturally lends itself to Speculative Decoding (SD) (Leviathan et al., 2023; Li et al., 2024; Cai et al., 2024; Liu et al., 2024; Gloeckle et al., 2024; Li et al., 2025b; Sun et al., 2025) by producing highly compatible token sequences. Since the second-stage *Thinking with Seed* generation is explicitly conditioned on the first-stage seed thoughts, the outputs exhibit strong token-level alignment — a critical factor for speculative acceleration. As shown in Figure 1, the *Thinking with Seed* trajectory closely follows the reasoning path established by the seed, resulting in higher token acceptance rates than standard auto-regressive drafting or corpus-based suffix matching.

**Our contributions are as follows.**

- We conduct a systematic study of how the granularity of seed thoughts affects the performance of LRMs.
- We propose a difficulty-aware seed-thought generation framework that adaptively selects granularity based on problem difficulty.
- To the best of our knowledge, we are the first to integrate *Thinking* and *No-Thinking* mode with speculative decoding, achieving dual-path speedups — shorter deliberative traces and faster token generation enabled by reusing seed thoughts as a model-free speculative corpus — within a single cohesive framework.
- Our experiments show up to $3.1\times$ end-to-end acceleration and 45.0% reduction in generation tokens with only 2.4% accuracy degradation on the MATH500 dataset on Qwen3 models, delivering superior efficiency–accuracy trade-offs compared to existing methods and advancing the state of the art. Seed thoughts effectively warm-up speculative decoding corpora, obtaining 16% additional reasoning acceleration than the hybrid SAM with EAGLE3.

## 2 RELATED WORKS

**Chain-of-Thought and Reasoning Models.** Chain-of-Thought (CoT) reasoning denotes a paradigm for enhancing the transparency and accuracy of large language models (LLMs) in complex reasoning tasks by explicitly generating intermediate logical steps that culminate in a final output (Wei et al., 2022b). Early foundational works (Wei et al., 2022b; Kojima et al., 2022) demonstrated that CoT prompting — either via few-shot examples or zero-shot instructions — significantly improves LLM performance on mathematical and logical benchmarks. After that, diverse CoT variants, including structured CoT (Li et al., 2023), ToT (Yao et al., 2023a), self-reflection (Ji et al., 2023), and self-correction (Huang et al., 2023), are proposed to enhance traceability and accuracy.

Chain-of-Thought and its variants guide LLMs to learn from in-context prompts and generate internal reasoning before generating the final answers. These works establish the cornerstones of recent reasoning models, agent, and Agentic AI, where models are trained to obtain the intrinsic capability of generating CoTs (Yao et al., 2023b; Jaech et al., 2024; Muennighoff et al., 2025; Guo et al., 2025; Yang et al., 2025). However, the initial solution or guidance generated during thinking may be inefficient and ineffective, resulting in overthinking and underthinking (Cuadron et al., 2025).

**Efficient Reasoning.** Recently, many training-free or training-based methods are proposed to reduce the number of generated tokens, improving the efficiency of test-time scaling. Auto-thinking utilizes external classifiers or LRMs' intrinsic task complexity classification capabilities to automatically switch between fast and slow thinking (Zhang et al., 2025c; Xing et al.). Self-evaluation methods prompt LRMs to assess their own confidence and decide when to stop reasoning (Wang et al., 2025c). Model merging directly merge LLM weights to trade-off the reasoning capability and cost (Team et al., 2025; Wu et al., 2025a). Multi-model frameworks use auxiliary evaluation mod-

els, reward models, or thought proposers to guide decoding (Fu et al., 2024; Kuhn et al., 2023; Sun et al., 2024). There are also concurrent works on improve LRM efficiency by generating thoughts with smaller reasoning models for speculative reasoning (Pan et al., 2025; Wang et al., 2025b;d). CoThink (Fan et al., 2025) leverages an instruction model to guide reasoning and reduce reasoning steps. Supervised Fine-Tuning (SFT) or Reinforcement Learning (RL) based methods train LRMs to with compact reasoning trajectories or length-aware rewards, respectively (Chen et al., 2024; Hou et al., 2025; Xia et al., 2025; Zhang et al., 2025a; Chen et al., 2025; Fang et al., 2025).

**Speculative Decoding.** Due to the lossless advantage, speculative decoding emerged as a widely studied latency mitigation paradigm by introducing a "draft-verify" pipeline: a lightweight draft model proposes candidate token sequences, and the target LLM verifies these candidates in parallel, accepting valid tokens to reduce sequential computation(Leviathan et al., 2023; Chen et al., 2023). After that, n-gram based drafter (He et al., 2023; Zhao et al., 2024; Luo et al., 2024; Hu et al., 2024; Oliaro et al., 2025), and auxiliary lightweight trained model(Li et al., 2024; 2025b; Cai et al., 2024; Liu et al., 2024; Gloeckle et al., 2024) based methods, infrastructure optimization (Miao et al., 2024) are proposed to improve the practical inference. Sun et al. (2025) systemically compared different existing speculative decoding methods for reasoning acceleration. Recently, He et al. (2025) proposed to accelerate RL training with SD enhanced with rollout responses tokens in adjacent training epochs.

Our work differs from prior studies in two key respects: 1) We focus on the synergy between Thinking and No-Thinking modes, rather than simply switching between them. Seed thoughts are generated in the No-Thinking mode and capture critical constraints and pivot **reasoning anchors** of the searching trajectories. They serve as additional hints that improve LRM confidence, speedup search convergence, and guide effective and efficient thinking, rather than searching from scratch. 2) We study how to leverage additional contexts, including seed thoughts, to warm-up speculative decoding corpora to further accelerate LRMs.

## 3 PRELIMINARY AND OBSERVATIONS

### 3.1 BACKGROUND AND NOTATIONS

To ground our setting, we first fix notation and delineate the generation paradigms considered in this work. Let $M$ denote a standard LRM. Given an input question $q$, the model generates a response sequence $r$. Two special tokens, $\langle\text{BOT}\rangle$ and $\langle\text{EOT}\rangle$, are used to demarcate the beginning and end of the explicit reasoning block.

**Thinking Mode.** The model generates a full chain-of-thought reasoning process before producing the final answer. The generation can be formally described as:

$$r_{\text{thinking}} = M(q + \langle\text{BOT}\rangle) = \langle\text{BOT}\rangle[\text{Thoughts}]\langle\text{EOT}\rangle[\text{Conclusion}]$$

**No-Thinking Mode.** The model is prompted to bypass the explicit reasoning process and generate the final answer directly. This is achieved by prepending a specific instruction inside the reasoning block that forces an immediate conclusion (Ma et al., 2025). The generation is:

$$r_{\text{nothinking}} = M(q + \langle\text{BOT}\rangle + s_{\text{skip}} + \langle\text{EOT}\rangle) = \langle\text{BOT}\rangle s_{\text{skip}}\langle\text{EOT}\rangle[\text{Conclusion}]$$

where $s_{\text{skip}}$ is a fixed string (e.g., "Okay, I think I have finished thinking.") that signals the model to skip deliberative thinking .

**Thinking with Seed.** To mitigate the inefficiency and path-dependence inherent in the standard *Thinking* mode, we introduce a hybrid generation strategy termed *Thinking with Seed*. This mode leverages a concise, high-quality seed thought $s_{seed}$ to initialize and guide the subsequent deliberative reasoning process. The seed $s_{seed}$ acts as structured scaffolding that shapes the subsequent reasoning trajectory by providing an initial sketch of the solution path. The model is then prompted to generate a full reasoning trajectory, explicitly conditioned on the seed $s$. The generation is formalized as:

$$r_{\text{SeedThink}} = M(q + s_{seed} + \langle\text{BOT}\rangle) = \langle\text{BOT}\rangle[\text{Extended Thoughts}]\langle\text{EOT}\rangle[\text{Conclusion}]$$

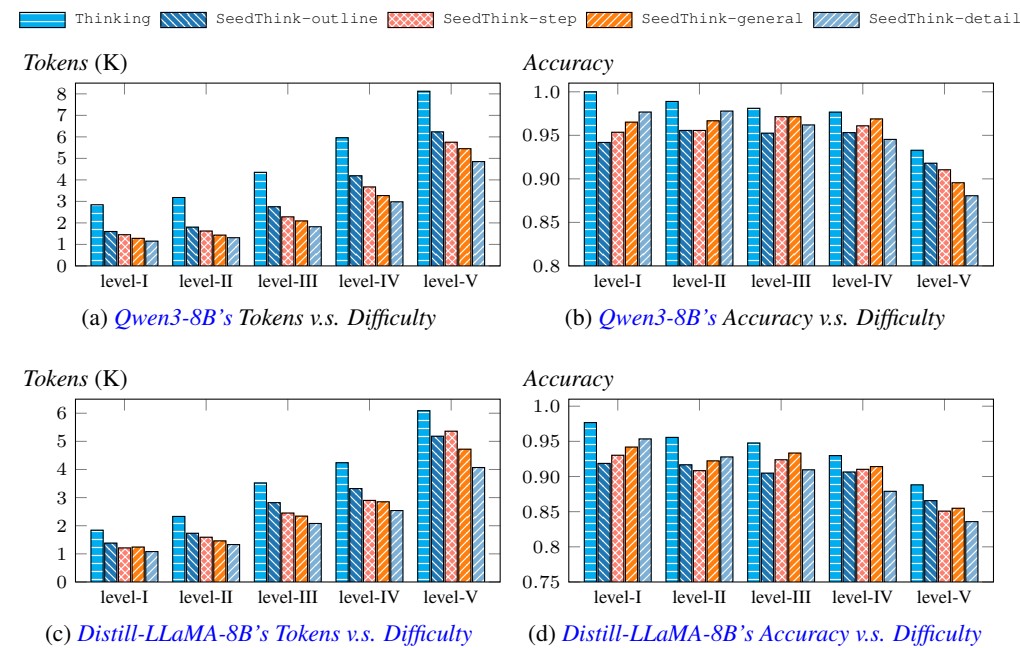

Figure 2: Comparison of Thinking mode and four different Thinking with Seed Thoughts (Seed-Think) variants using Qwen3-8B and DeepSeek-R1-Distill-LLaMA-8B across different difficulty levels of MATH500 dataset.

## 3.2 OBSERVATION

In this section, we investigate how the level of detail in seed thoughts affects the reasoning trajectories of LRMs, revealing opportunities for optimization in LRMs' reasoning efficiency.

**Experimental settings.** We conduct a pilot study using the MATH500 datasets (Hendrycks et al., 2021) to analyze the impact of seed granularity on the performance of two widely used LRMs: Qwen3-8B and DeepSeek-R1-Distill-LLaMA-8B. We define four levels of seed granularity:

- *Outline seed:* High-level strategic breakdown focusing only on outline without numbers, formulas, or calculations

- *Key-steps seed:* Main reasoning steps with important formulas/concepts mentioned.

- *General-approach seed:* Step-by-step plan with brief explanations, including formulas and concepts, but avoiding extensive calculations

- *Detailed-Steps seed:* Near-complete solution with formulas, initial setups, and simple calculations, leaving only complex reasoning for completion

We compare these against the baseline *Thinking* approach (no seed). For each problem, we generate seeds at each granularity level using specialized prompts. More details about parameter settings and prompts can be found in Appendix B.

**Results and analysis.** Our pilot study yields a clear central finding: the optimal granularity of a seed thought follows a non-monotonic relationship with problem difficulty. As illustrated in Figure 2, we observe a striking pattern: on the simplest problems (level-I/II), the most detailed seeds achieve the highest accuracy alongside the greatest efficiency gains; for problems of intermediate difficulty (level-III/IV), a balanced general approach seed proves most accurate; while on the hardest problems (level-V), high-level outline seeds become most robust, outperforming detailed alternatives.

This tri-phasic pattern highlights a sophisticated trade-off between **guidance** and **flexibility**. On simple problems, a highly specific seed provides strong, beneficial constraints that focus the search without causing harm. At medium difficulty, an intermediate level of guidance offers the ideal balance of direction and flexibility. For the most complex problems, however, high-level strategic scaffolding becomes essential to avoid leading the model down incorrect, overly constrained paths.

Figure 3: The SeedThink framework comprises three core modules: (1) Difficulty Estimator: Assesses problem complexity; (2) Seed Thought Generator: Produces a difficulty-aware seed solution; (3) Seed-Guided Reasoning: Utilizes the seed for focused reasoning and as a high-quality draft for speculative decoding, achieving dual-path acceleration.

The universal efficiency gains of SeedThink (**23%-59%** token reduction at all levels) confirm its value for efficient test-time control. The dramatic shift in the relative performance of seed types across the difficulty spectrum, however, reveals that a static seeding strategy is fundamentally suboptimal. This finding directly motivates our subsequent proposal of a difficulty-aware seeding policy, which dynamically modulates seed granularity to track this shifting optimum, as detailed in the next section.

## 4 METHOD

As visualized in Figure 3, SeedThink improves reasoning efficiency by proposing difficulty-aware seed thoughts as additional thinking hints and token corpora to improve thinking generation confidence and enhance speculative decoding acceptance rate, respectively. The framework consists of three modules: 1) Difficulty estimator for more accurate alignment with problem difficulties in Section 4.2, 2) Seed thought generator, which generate suitable seed granularity based on the previous prompt difficulty estimation, in Section 4.3, 3) Seed guided reasoning with enhanced speculative decoding for shorter and faster test-time scaling in Section 4.4.

### 4.1 PROBLEM FORMULATION

Our SeedThink framework involves a two-stage generation process: (1) generating a seed thought with appropriate granularity, and (2) producing the final reasoning trajectory conditioned on the seed as formulated in Section 3.1. Thus, the key to this process is selecting the suitable seed granularity for diverse prompt difficulties to trade-off efficiency and performance.

We formulate the seed thought selection as an optimization problem that maximizes a composite objective function balancing efficiency against accuracy. Given a problem $q$ with estimated difficulty $d$, we select the optimal seed granularity $g^* \in \mathcal{G}$ from our predefined seed set.

Formally, for a problem with difficulty $d$, the ideal seed granularity $g^* \in \mathcal{G}$ should maximize:

$$g^*(d) = \arg\max_{g \in \mathcal{G}} \left[ \mathcal{E}(g, d) + \lambda \cdot \mathcal{A}(g, d) \right]$$

where $\mathcal{E}(g, d) = \frac{T_{\text{think}}(d) - T_g(d)}{T_{\text{think}}(d)}$ represents relative efficiency gain, $\mathcal{A}(g, d) = \frac{A_g(d) - A_{\text{think}}(d)}{A_{\text{think}}(d)}$ represents relative accuracy change, and $\lambda \geq 0$ is a trade-off coefficient. Here, $T_g(d)$ and $A_g(d)$ are the number of tokens and accuracy under granularity $g$ at difficulty $d$, while $T_{\text{think}}(d)$ and $A_{\text{think}}(d)$ are the corresponding quantities under the original *Thinking* mode.

## 4.2 DIFFICULTY ESTIMATOR

Existing research from Liu et al. (2025) indicates that LRMs fundamentally lack the capacity for accurate, autonomous difficulty cognition. Equipping models with this metacognitive ability typically necessitates carefully designed SFT or RL, which entail significant computational expense. For the sake of efficiency, we avoid this training overhead by introducing a separate difficulty estimator as a crucial first step in our methodology. Accordingly, we frame the problem of difficulty assessment as a text classification task that is decoupled from the LRM's own reasoning process. The specific architecture, detailed training procedure, and an evaluation of the estimator's effectiveness are provided in Appendix D. The difficulty estimator module assesses the complexity level of input problems to determine the appropriate seeding strategy. Formally, it maps a problem $q$ to a difficulty level $d \in \mathcal{D}$, where $\mathcal{D} = \{\text{Simple}, \text{Moderate}, \text{Hard}\}$ is the set of difficulty categories.

## 4.3 SEED THOUGHT GENERATOR

Based on our empirical findings in Section 3.2, we propose a deterministic seeding policy $\pi(d)$ that directly maps difficulty to seed granularity:

$$\pi(d) = \begin{cases} \text{detailed steps} & \text{if } d = \text{Simple} \\ \text{key steps} & \text{if } d = \text{Moderate} \\ \text{outline} & \text{if } d = \text{Hard} \end{cases}$$

The seed generator then produces:

$$Seed_{\pi(d)} = M(\mathcal{T}_{\pi(d)}(q)) \tag{1}$$

where $\mathcal{T}_g$ is the prompt template for granularity $g$.

## 4.4 SEED GUIDED REASONING WITH ENHANCED SPECULATIVE DECODING

Building upon the difficulty-aware generation of the seed thought $Seed_{\pi(d)}$, the seed guided reasoning process produces the final output by explicitly conditioning on the seed:

$$r_{\text{SeedThink}} = M(q, Seed_{\pi(d)}) \tag{2}$$

The sequential dependency between Eq. (1) and Eq. (2) is the cornerstone of our method. The final reasoning output $r_{\text{SeedThink}}$ is explicitly conditioned on the $Seed_{\pi(d)}$, creating a strong **path dependence**. Combined with the **model consistency** of using the same model $M$ for both generations, this design ensures a high degree of token-level alignment between the two sequences. It is this inherent alignment that makes the seed from Eq. (1) a naturally high-quality draft for speculative decoding, as the verification of the seed tokens during the generation of Eq. (2) achieves higher acceptance rates than drafts lacking such a causal link. Thus, the seed thought serves a dual purpose: it structures the reasoning path deterministically, while its token sequence functions as an internal, optimized draft for accelerated generation.

We formally analyze why this conditioning leads to high token-level alignment in Appendix F, providing theoretical guarantees for our method's efficiency.

# 5 EVALUATION

## 5.1 EXPERIMENTAL SETUP

**Datasets.** We evaluate on three math datasets of increasing difficulty: **GSM8K**(1,319 grade-school problems) (Cobbe et al., 2021), **MATH500** (a 500-problem subset of the MATH benchmark containing high-school competition problems) (Hendrycks et al., 2021), and **AIME 2024** (30 challenging Olympiad-level problems). We report solution accuracy, wall time, and the average number of generated tokens (response length) as key metrics. We report accuracy as Pass@1 (percentage of problems solved correctly by the first sample). All generated answers are checked using Hugging-Face's official Math-Verify[1] for fairness and reproducibility. Due to the small size of AIME 2024,

---

[1]https://github.com/huggingface/Math-Verify

Table 1: Accuracy, average response length (Length), and average runtime of different methods on three math benchmarks. The best and second results are bolded and underlined, respectively. For ease of comparison, the results include absolute accuracy differences, speedup ratios relative to the Think baseline, and relative length differences.

| Method | GSM8K | | | MATH 500 | | | AIME 2024 | | |
| --- | --- | --- | --- | --- | --- | --- | --- | --- | --- |
| | Accuracy ↑ | Runtime (s) ↓ | Length ↓ | Accuracy ↑ | Runtime (s) ↓ | Length ↓ | Accuracy ↑ | Runtime (s) ↓ | Length ↓ |
| *DeepSeek-R1-Distill-LLaMA-8B* | | | | | | | | | |
| *Think* | 76.50% | 113.9 | 4029 | 87.60% | 104.6 | 3627 | 46.67% | 299.1 | 11288 |
| *NoThink* | 76.80% +0.3% | 12.0 9.5× | 320 -92.1% | 66.20% -21.4% | 19.7 5.3× | 682 -81.2% | 10.00% -36.7% | 60.6 4.9× | 1917 -83.0% |
| *JointThink* | 80.12% +3.6% | 143.2 0.8× | 4968 +23.3% | 90.20% +2.6% | 144.2 0.7× | 5018 +24.5% | 43.33% -3.3% | 478.0 0.6× | 15220 +34.8% |
| *CoThink* | 77.64% +1.1% | 98.1 1.2× | 3433 -14.8% | 81.20% -6.4% | 85.7 1.2× | 2890 -19.9% | 40.00% -6.6% | 233.0 1.3× | 8630 -23.5% |
| **SeedThink (ours)** | 78.22% +1.7% | 61.3 1.9× | 3647 -9.5% | 82.80% -4.8% | 45.1 2.3× | 2611 -28.0% | 43.33% -3.3% | 131.3 2.3× | 9049 -19.9% |
| *Qwen3-8B* | | | | | | | | | |
| *Think* | 95.00% | 94.6 | 2405 | 96.20% | 191.4 | 5432 | 64.16% | 358.6 | 11863 |
| *NoThink* | 90.22% -4.8% | 10.3 9.2× | 956 -89.0% | 83.80% -12.4% | 36.2 5.3× | 956 -82.4% | 26.67% -37.5% | 116.1 3.1× | 3634 -69.4% |
| *JointThink* | 95.83% +0.8% | 109.4 0.9× | 2783 +15.7% | 97.00% 0.8% | 251.4 0.8× | 7040 +29.6% | 60.00% -4.16% | 568.1 0.6× | 17823 +50.2% |
| *CoThink* | 94.62% -0.4% | 37.6 2.5× | 865 -64.0% | 94.20% -2.0% | 146.4 1.3× | 4015 -26.1% | 63.33% -0.8% | 401.6 0.9× | 12730 +7.3% |
| **SeedThink (ours)** | 94.84% -0.2% | 24.9 3.8× | 801 -66.7% | 94.40% -1.8% | 72.2 2.7× | 3630 -33.2% | 61.16% -3.0% | 176.4 2.0× | 10268 -13.5% |
| *Qwen3-14B* | | | | | | | | | |
| *Think* | 95.53% | 75.5 | 1881 | 96.80% | 176.9 | 4741 | 66.67% | 395.6 | 11291 |
| *NoThink* | 90.85% -4.7% | 11.9 6.3× | 277 -85.3% | 86.40% -10.4% | 34.5 5.1× | 896 -81.1% | 33.33% -33.3% | 159.5 2.5× | 4453 -60.6% |
| *JointThink* | 96.12% +0.6% | 96.7 0.8× | 2388 +27.0% | 97.40% +0.6% | 232.3 0.8× | 6076 +28.2% | 66.67% -0.0% | 611.4 0.6× | 17057 +51.1% |
| *CoThink* | 93.86% -1.7% | 28.2 2.7× | 680 -63.8% | 93.80% -3.0% | 114.5 1.5× | 2959 -38.0% | 72.50% +5.83% | 350.5 1.1× | 9879 -12.5% |
| **SeedThink (ours)** | 94.39% -1.1% | 18.4 4.1× | 588 -68.7% | 94.60% -2.2% | 60.7 2.9× | 2761 -42.0% | 70.83% +4.16% | 188.5 2.1× | 8744 -22.4% |
| *Qwen3-32B* | | | | | | | | | |
| *Think* | 95.30% | 101.6 | 1731 | 97.20% | 282.6 | 4657 | 75.00% | 836.3 | 12571 |
| *NoThink* | 91.52% -3.8% | 18.4 5.5× | 286 -83.5% | 86.40% -10.8% | 46.9 6.0× | 774 -83.4% | 26.67% -48.3% | 170.9 4.9× | 2769 -78.0% |
| *JointThink* | 96.41% +1.1% | 123.6 0.8× | 2099 +21.3% | 97.60% +0.4% | 358.3 0.8× | 5924 +27.2% | 75.00% -0.0% | 1267.9 0.7× | 19337 +53.8% |
| *CoThink* | 94.47% -0.8% | 42.3 2.4× | 620 -64.2% | 94.40% -2.8% | 163.0 1.7× | 2891 -38.0% | 73.33% -1.7% | 588.6 1.4× | 9416 -25.1% |
| **SeedThink (ours)** | 93.93% -1.4% | 24.6 4.1× | 560 -67.6% | 94.80% -2.4% | 90.1 3.1× | 2559 -45.0% | 72.50% -2.5% | 301.7 2.8× | 8408 -33.1% |

we generate 4 responses per problem and report the average performance. For all models, we use a 16K context window. Following the recommendations to balance diversity and coherence in the official model cards, we set the temperature to 0.6 for *Thinking* mode and 0.7 for *No-Thinking* mode, respectively.

**Models.** We conduct our main experiments on four publicly available Large Reasoning Models (LRMs) of varying scales: DeepSeek-R1-Distill-Llama-8B (Guo et al., 2025), Qwen3-8B, Qwen3-14B, and Qwen3-32B (Yang et al., 2025). These models are specifically designed for complex reasoning and Qwen3 natively support both *Thinking* and *No-Thinking* generation modes, making them ideal testbeds for our proposed efficiency improvements.

**Baselines.** We compare our approach against four representative baseline methods:

- *Think:* The standard *Thinking* mode, where the model generates a full, multi-step reasoning trace before producing a final answer. This represents the high-cost, high-performance upper bound for LRMs.
- *NoThink:* The standard *No-Thinking* mode, where the model generates a direct, concise answer without an explicit reasoning trace. This represents the low-cost, lower-performance baseline.
- *JointThink (Wu et al., 2025b):* A calibration-based method that first generates answers in parallel using both *Thinking* and *No-Thinking* modes. If the answers from two modes are inconsistent, it triggers a second *Thinking* round conditioned on both candidates' answers to improve robustness.
- *CoThink (Fan et al., 2025):* A two-stage pipeline where an Instruct model first creates a very short reasoning outline, which is then refined by a reasoning model.

## 5.2 MAIN RESULTS

**Superior Efficiency-Effectiveness Trade-off.** As shown in Table 1, our proposed SeedThink method achieves a remarkable balance between computational efficiency and reasoning accuracy across all model scales and benchmarks, achieving up to $4.1\times$ reasoning acceleration. On the challenging MATH 500 dataset, SeedThink maintains competitive accuracy while accelerating inference by $2.3 \sim 3.1\times$ and reducing token length by 28.0%-45.0% compared to standard Thinking mode, outperforming the state-of-the-art (SOTA) baseline CoThink (Fan et al., 2025) in all three metrics. While CoThink utilizes only coarse-grain outlines, which proves sub-optimal across diverse difficulty levels, our method dynamically selects seed granularity based on the problem difficulty. This adaptability allows SeedThink to provide detailed guidance for simple problems where specificity is

beneficial, while maintaining the flexibility of high-level outlines for complex problems where rigid guidance may be detrimental.

The advantages of SeedThink become particularly evident when compared to alternative efficiency-oriented methods. While JointThink achieves modest accuracy improvements of 0.4%-2.6% on MATH500, it leads to $20 \sim 30\%$ slowdown and requires $24 \sim 29\%$ longer reasoning trajectories. In contrast, SeedThink delivers substantially greater efficiency gains while maintaining accuracy loss within 2.4% of the standard Thinking baseline. On the more challenging AIME 2024 benchmark, SeedThink consistently outperforms both alternatives, often matching or even improving accuracy while improving efficiency more than $2\times$ across all model scales. These results demonstrate that SeedThink achieves a superior efficiency-effectiveness trade-off, providing the practical benefits of dramatically reduced computational costs while preserving the reasoning quality of LRMs.

**Dual-path speedups.** SeedThink achieves up to $4.1\times$ acceleration through two complementary mechanisms: (1) reduced reasoning trajectory length via strategic seed thought guidance, and (2) speculative decoding acceleration enabled by high token-level alignment between seed thoughts and final reasoning paths. The $9.5 \sim 68.8\%$ token length reductions directly translate to proportionally fewer decoding steps, while the even greater speedup of $1.9 \sim 4.1\times$ indicates additional gains from the reused seed thought as high-quality drafts for speculative decoding. We further conduct an ablation study in Section 5.3 about the effect of reusing them for warming up SD corpora.

## 5.3 ABLATION STUDY ON SPECULATIVE DECODING

To validate the core hypothesis from Section 4.4—that the **path dependence** between seed generation and final reasoning creates superior token-level alignment for speculative decoding—we conduct a carefully designed ablation study comparing three distinct paradigms.

First, we reproduce existing speculative decoding approaches: **Think + SAM** (Hu et al., 2024) and **Think + SAM + Eagle3** (Li et al., 2025b), which represent SOTA model-free and model-based methods applied to standard reasoning trajectories (Sun et al., 2025). Second, we test a warm-up setting where seeds serve only as external draft corpora for SAM, isolating the pure speculative utility of seeds from our path-dependent generation paradigm in Eq. (2).

Table 2 reveals a clear but modest distinction: while the warm-up configuration brings gains (1.45× and 2.08× speedups), these are consistently lower than SeedThink's 1.56× and 2.24× speedups with the same components.

This performance gap provides direct evidence for our theoretical claim: the sequential dependence between Eqs. (1) and (2) is the cornerstone of our method's effectiveness. When seed thoughts are used only as external corpora for warm-up, MAT gains remain limited. In contrast, **explicitly conditioning final reasoning on seed thoughts creates genuine path dependence and yields measurably higher MAT.**

In addition, we investigated how the length of the seed affects speculative performance. The corresponding experimental results and analysis are provided in the appendix E.

| Method | MAT | Throughput | Speed up |
|---|---|---|---|
| Think | 1.00 | 28.4 | – |
|   + SAM | 1.75 | 38.8 | 1.37× |
|   + SAM + w | 1.88 | 41.2 | 1.45× |
|   + SAM + Eagle3 | 3.26 | 54.5 | 1.92× |
|   + all components | 3.41 | 59.0 | 2.08× |
| SeedThink | 1.00 | 28.6 | 1.00× |
|   + SAM | 1.94 | 44.2 | 1.56× |
|   + SAM + Eagle3 | 3.65 | 63.6 | 2.24× |

Table 2: Ablation study on MATH500 using Qwen3-4B, comparing speculative decoding performance. "w" denotes warm-up. MAT: Mean Accepted Tokens; Throughput: tokens per second. Speed up are normalized to baseline Thinking.

The results conclusively demonstrate that it is not merely about supplying extra corpora from seed thoughts, but their **integrated role in the generation process** that enables superior speculative performance.

Our two-stage framework creates a cohesive reasoning trajectory where each step naturally follows from the previous, producing the high-quality token alignment that enhances both model-free and model-based speculative decoding approaches.

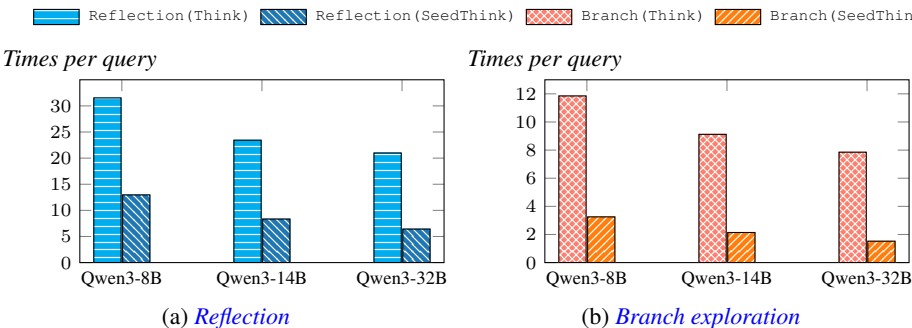

Figure 4: Comparison of Reflection token and Branch exploration tokens of Think and SeedThink on Qwen3-8B, Qwen3-14B, Qwen3-32B

## 5.4 ANALYSIS OF REASONING TRAJECTORY OPTIMIZATION

The inefficiency of current LRMs is largely driven by two unproductive behaviors: *repetitive verification and redundant branching explorations*. These behaviors can be measured with the number of reflection and branch tokens, which are defined below:

**Reflection** represents self-verification behaviors where the model double-checks its reasoning through repetitive validation cycles. This manifests linguistically through phrases like "wait", "let me check", and "let me verify"—indicators of conservative reasoning patterns that often continue well after a solution has been adequately established (Wang et al., 2025a).

**Branch exploration** encompasses the model's tendency to consider alternative solution paths, signaled by terms like "alternatively", "another way", and "different approach". While potentially valuable for complex problem-solving, uncontrolled branching frequently leads to computational inefficiency as the model explores unpromising directions.

Our analysis reveals that SeedThink significantly reduces these two inefficient reasoning behaviors. As shown in Figure 4, for Qwen3-8B, reflection tokens decrease by 58.9% while branch tokens drop by 72.6%. This pattern holds consistently across model scales, with Qwen3-32B models showing the most dramatic reductions: 69.5% fewer reflection tokens and 80.6% fewer branch tokens.

The mechanism behind this optimization lies in the seed's role as a **reasoning anchor**. By providing high-quality initial guidance, SeedThink preemptively addresses the uncertainties that typically trigger verification cycles and alternative path exploration. The seed establishes a focused solution trajectory, reducing the need for extensive backtracking and minimizing unproductive branching.

## 6 CONCLUSIONS

In this work, we addressed the dual challenges of inefficiency and path dependence in large reasoning models (LRMs) that leverage test-time compute. Our proposed solution, SeedThink, fundamentally reframes the initialization of the reasoning process. By generating a concise, high-quality "seed thought" to guide the LRM, we steer its exploration toward productive regions of the search space. We demonstrated that the granularity of these seeds is critical and introduced a difficulty-aware seeding policy that adapts seed detail to problem complexity—using high-level seeds for hard problems to maintain flexibility and detailed seeds for easier ones to maximize efficiency.

This two-stage framework provides a dual-path acceleration. First, it shortens the reasoning trajectory by pruning unproductive branches, reducing token generation by up to 45.0%. Second, we showed for the first time that this paradigm naturally integrates with speculative decoding, using the seed thought as a highly quality draft. This synergy resulted in up to a 4.1× end-to-end speedup, achieving a state-of-the-art efficiency-accuracy trade-off. Our findings underscore the importance of guided initialization in complex, multi-step generation tasks. Future work could explore learning the optimal seeding policy directly, rather than relying on a difficulty classifier. Ultimately, by shaping the start of the journey, seed thoughts enable LRMs to reason more efficiently, making powerful deliberative reasoning more practical for real-world applications.

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

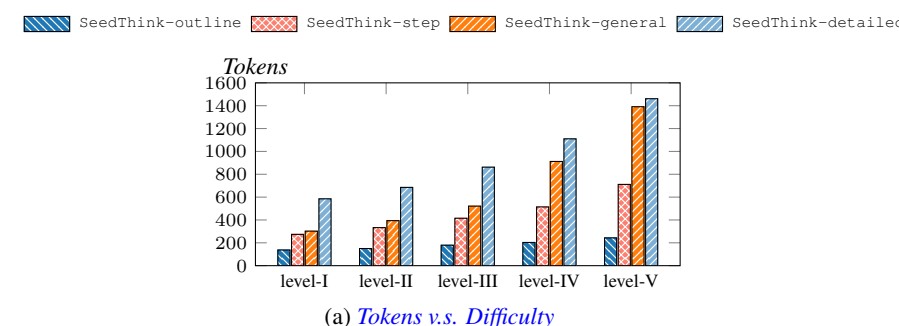

(a) *Tokens v.s. Difficulty*

Figure 5: Different seed prompt and it's generated seed length of Qwen3-14B on Math500 dataset.

## A   THE USE OF LARGE LANGUAGE MODELS (LLMs)

Large Language Models (LLMs) were employed solely for the purpose of enhancing the linguistic clarity and stylistic refinement of this manuscript.

## B   SUPPLEMENTARY MATERIALS FOR SECTION 3.2

As described in Section 3.2, we conduct a pilot study using the MATH500 datasets to analyze the impact of seed granularity on the performance of two widely used LRMs Qwen3-8B and DeepSeek-R1-Distill-LLaMA-8B. For seed thought generation, we use the No-Thinking mode with four different prompt to control the seed granularity. For the second round of generation, we use the thinking mode while providing the seed as an additional reference through the thinking with seed prompt. Both the seed generation prompt and the thinking with seed prompt can be found in C.

We use an 8K context window for seed thought generation and 16K context window for thinking with seed reasoning process. Following the recommendations to balance diversity and coherence in the official model cards, we set the temperature to 0.6 for Thinking mode and 0.7 for No-Thinking mode,respectively.

The generated seed length of four different SeedThink variants are reported in 5.

## C   DETAILS ON PROMPT

**Outline seed generation prompt**

```
<|User|>You are a reasoning assistant.  Your job is to break
down a complex problem into 2 to 4 high-level reasoning steps.
Focus only on outlining the general approach or strategy.  Do not
include any numbers, formulas, or final answers.  Avoid specific
calculations or details|only describe the logic behind solving the
problem.  Please break down the following problem.  [question]
<|Assistant|><think></think> \n
```

**key steps seed generation prompt**

```
<|User|>You are a reasoning assistant.  Your task is to rapidly
provide key steps for solving the given math problem.  Include the
main reasoning steps without performing detailed calculations.
Mention any important formulas or concepts needed, but do not
compute numerical results or give the final answer.  [question]
<|Assistant|><think></think> \n
```

**General approach seed generation prompt**

```
<|User|>You are a reasoning assistant.  Quickly provide a general
approach to solve the given math problem.  Include a step-by-step
plan with brief explanations of each step.  You may mention
formulas and concepts, but avoid extensive calculations.  The goal
is to set up the problem for detailed reasoning later.  [question]
<|Assistant|><think></think> \n
```

**Detailed steps seed generation prompt**

```
<|User|>You are a reasoning assistant.  Your task is to promptly
provide detailed steps for solving the given math problem.  Include
necessary formulas, initial setups, and perform simple calculations
if needed.  However, leave the most complex calculations or
reasoning for the next stage.  Provide as much detail as possible
without giving the final answer.  [question]
<|Assistant|><think></think> \n
```

**Thinking with seed generation prompt**

```
<|User|> Reference Seed solution from a instruct model:
[Reference answer]
It is crucial to critically evaluate the information provided in
the reference, recognizing that it may be biased or incorrect.  If
you think the reference is incorrect, try to correct it to become
your answer.
[question]
<|Assistant|><think>\n
```

# D  DIFFICULTY ESTIMATOR AS A TEXT CLASSIFICATION TASK

To implement this classifier, we employ a RoBERTa model (Liu et al., 2019). This choice is motivated by RoBERTa's well-established and outstanding performance on a wide range of text classification benchmarks, where it consistently delivers robust and accurate predictions. Furthermore, compared to the large reasoning models that are the focus of our study, a RoBERTa-based classifier is relatively lightweight, allowing for efficient inference without introducing significant computational burden. This combination of high classification accuracy and operational efficiency makes it an ideal candidate for our difficulty estimation step.

We formulate difficulty estimation as a 3-class classification task. Specifically, we train the model on a mathematical dataset comprising 12,500 samples with inherent difficulty annotations. The original fine-grained labels (on a scale of 1-5) are mapped to three broader categories: labels 1-2 are grouped as *easy*, labels 3-4 as *moderate*, and label 5 as *hard*. This categorization captures meaningful distinctions in problem complexity while maintaining sufficient data for each class.

Our implementation uses the RoBERTa-base architecture, which contains 125 million parameters configured with 12 transformer layers, a hidden state size of 768, and 12 attention heads. The model is trained for 10 epochs with a batch size of 16. This demonstrates the model's strong capability in reliably assessing problem difficulty.

The analysis below provide a comprehensive breakdown of the estimator's performance on the Math500 dataset. It shows that even when the estimator misclassifies a problem, it does not lead to a catastrophic failure. Instead, it results in a predictable and often acceptable trade-off between computational cost and solution accuracy. As the table3 shows, we can analyze the two distinct types of misclassification scenarios.

The first type occurs when a **harder problem is misclassified as easier** (e.g., a True Hard problem predicted as Medium). In this case, the model receives a seed that is more detailed than opti-

Table 3: A comprehensive breakdown of the estimator's performance on the Math500 dataset.

| Label | Predicted: Easy | | | Predicted: Medium | | | Predicted: Hard | | |
|---|---|---|---|---|---|---|---|---|---|
| | Cnt. | Acc. | Length | Cnt. | Acc. | Length | Cnt. | Acc. | Length |
| **Easy** | **125** | 96.8% | 1,206 | 7 | 100.0% | 1,676 | 1 | 100.0% | 2,853 |
| **Medium** | 30 | 90.0% | 1,950 | **191** | 97.9% | 2,067 | 12 | 100.0% | 2,222 |
| **Hard** | 2 | 50.0% | 3,105 | 26 | 84.6% | 3,705 | **106** | 92.5% | 5,257 |

mal. While this can lead to a decrease in accuracy—for instance, 'Hard' problems misclassified as 'Medium' see their accuracy drop from 92.5% to 84.6%—this is coupled with a **corresponding reduction in generation length**. The average generation length for these same problems plummets from **5258 to 3705 tokens**. This outcome represents a clear trade-off: a modest reduction in accuracy for a substantial improvement in computational speed.

The second type of error is when an **easier problem is misclassified as more difficult** (e.g., a True Easy problem predicted as Medium). Here, the model is given a higher-level, more abstract seed than necessary. This prompts a more cautious reasoning process, which our data shows **actually increases accuracy**. All those misclassified types achieve **100% accuracy**, an improvement over the 96.8% for those classified correctly. This gain in robustness comes at the expected cost of efficiency, with the average token length increasing from 1206 to 1676. This scenario again illustrates a trade-off: sacrificing efficiency for a gain in accuracy.

In summary, The vast majority of samples (422/500, or 84.4%) are classified correctly, achieving a high accuracy of **96.2%**. On the 78 misclassified instances, performance remains strong at **89.7%** accuracy.

# E    ABLATION STUDY ON SEED LENGTH

To investigate whether longer or more abstract seeds lead to saturation in speculative gains, we conducted additional experiments. As shown in Table 4, rather than observing saturation, we find that **longer seeds generally increase speculative gains** (speed up from **1.47 to 1.64** and MAT from **1.65 to 1.96**), as they produce more opportunities for suffix matches. **We do not observe saturation within the studied range**, and we note that seed length is intentionally constrained: excessively long seeds would undermine the two-stage pipeline's efficiency. Within this realistic range, speculative gains (Mean accept tokens) grow monotonically.

Table 4: Speculative gains (using SAM decoding) of different seed prompt Qwen3-14B on the Math500 dataset. MAT short for Mean Accept Token.

| Seeds | MAT | Throughput (tokens/s) | Speed up from SD | Length | Accuracy |
|---|---|---|---|---|---|
| Think (w/o SAM) | 0 | 28.44 | – | 4,741 | 96.8% |
| Think (w/ SAM) | 1.65 | 40.26 | × 1.42 | 4,776 | 96.8% |
| SeedThink (outline seed) | 1.73 | 41.72 | × 1.47 | 2,959 | 93.8% |
| SeedThink (key_steps seed) | 1.82 | 43.88 | × 1.54 | 2,847 | 93.4% |
| SeedThink (general_approach seed) | 1.88 | 45.04 | × 1.58 | 2,735 | 95.6% |
| SeedThink (detailed_steps seed) | 1.96 | 46.87 | × 1.64 | 2,650 | 94.4% |

# F    TOKEN-LEVEL ALIGNMENT: $n$-GRAM PRESERVATION UNDER SEED-GUIDED REASONING

We provide a theoretical justification for why $n$-grams appearing in the seed sequence tend to reappear with high probability in the final reasoning trace. The key assumption is that the change of prompt template between seed generation and reasoning introduces only a small perturbation in the model's input representation.

## F.1 PRELIMINARIES

Let the seed sequence be

$$s = (s_1, s_2, \ldots, s_k),$$

generated by the model $M$ under a seed template. The final reasoning sequence is

$$r = (r_1, r_2, \ldots),$$

generated by the same model $M$ but conditioned on $(q, s)$.

For any prefix $u$, denote the next-token distributions by

$$p_s(\cdot \mid u), \quad p_r(\cdot \mid u),$$

corresponding respectively to the seed-generation context and the final-reasoning context. We define the stepwise total variation distance

$$\varepsilon_t(u) = \big\| p_s(\cdot \mid u) - p_r(\cdot \mid u) \big\|_{\mathrm{TV}}.$$

We further define its expectation with respect to prefixes encountered during reasoning:

$$\bar{\varepsilon}_t = \mathbb{E}_{r_{<t} \sim P_r}[\varepsilon_t(r_{<t})].$$

**Assumption (Template Stability).** The difference between the seed template and the reasoning template induces at most a small perturbation $\delta$ in the model's internal embeddings. Together with the Lipschitz continuity of the model's logit mapping and the stability of the softmax function, this implies the existence of a constant $\varepsilon$ such that

$$\bar{\varepsilon}_t \leq \varepsilon \quad \text{for all } t.$$

## F.2 MAIN LEMMA: $n$-GRAM ALIGNMENT AT MATCHED POSITIONS

**Lemma 1.** *For any $n$-gram $g = s_{i:i+n-1}$ appearing in the seed, the probability that the final reasoning sequence reproduces the same $n$-gram at the same position satisfies*

$$\Pr\big[r_{i:i+n-1} = g\big] \geq \prod_{t=i}^{i+n-1} \big(1 - \bar{\varepsilon}_t\big).$$

*Under the uniform bound $\bar{\varepsilon}_t \leq \varepsilon$, this becomes*

$$\Pr\big[r_{i:i+n-1} = g\big] \geq (1 - \varepsilon)^n \geq 1 - n\varepsilon.$$

*Proof.* Condition on the event that $r_{i:t-1} = s_{i:t-1}$. Under this condition, the total variation distance between $p_s(\cdot \mid r_{<t})$ and $p_r(\cdot \mid r_{<t})$ is at most $\varepsilon_t(r_{<t})$. By the optimal coupling characterization of total variation distance,

$$\Pr\big[r_t = s_t \mid r_{i:t-1} = s_{i:t-1}\big] \geq 1 - \varepsilon_t(r_{<t}).$$

Taking expectation over possible prefixes and applying the chain rule yields

$$\Pr\big[r_{i:i+n-1} = g\big] \geq \prod_{t=i}^{i+n-1} (1 - \bar{\varepsilon}_t).$$

The uniform bound follows immediately. $\qquad\square$

## F.3 COROLLARY: APPEARANCE AT ANY POSITION

Let $J \subseteq \{1, 2, \ldots, T\}$ be any candidate set of positions. By applying a union bound to Lemma 1, we obtain:

**Corollary 1.** *For any $n$-gram $g$ in the seed,*

$$\Pr\big[\exists j \in J : r_{j:j+n-1} = g\big] \geq 1 - \sum_{j \in J}\Big(1 - \prod_{t=j}^{j+n-1} (1 - \bar{\varepsilon}_t)\Big).$$

*Under the assumption $\bar{\varepsilon}_t \leq \varepsilon$,*

$$\Pr\big[\exists j \in J : r_{j:j+n-1} = g\big] \geq 1 - |J|\big(1 - (1 - \varepsilon)^n\big) \approx 1 - |J|n\varepsilon.$$

### F.4 INTERPRETATION

Under the Template Stability Assumption—which bounds the embedding perturbation $\delta$ between the seed-generation and reasoning-generation contexts— the resulting bound on stepwise total variation distance guarantees that short and medium-length $n$-grams in the seed are preserved with high probability in the model's subsequent reasoning. The alignment arises directly from the shared model parameters and the strong causal dependence between the seed and the final reasoning trace.

