# OpenReview forum: "SeedThink: Test-Time Control via Seed-Thought Initialization"
_ICLR.cc/2026/Conference — Submitted to ICLR 2026_

### Official Review · Reviewer_7GgU · 2025-10-28

**Soundness:** 2
**Presentation:** 2
**Contribution:** 3
**Rating:** 4
**Confidence:** 3

**Summary:**

SeedThink proposes a two-stage test-time reasoning method where a short “seed thought” is first generated to guide the model’s later detailed reasoning.  It combines a RoBERTa-based difficulty estimator, seed generation using the same LLM, and speculative decoding for early acceleration.  Experiments on Qwen3-8B show shorter reasoning trajectories and up to 2.2× speed-up.

**Strengths:**

1. Proposes a clear and novel two-stage reasoning framework that uses a self-generated seed to guide and structure test-time reasoning.
2. Demonstrates shorter, more focused reasoning trajectories with consistent accuracy and efficiency gains on reasoning benchmarks.
3. Integrates speculative decoding in a simple, training-free way to accelerate early decoding using the seed as a high-quality draft.

**Weaknesses:**

§3.2 Observation
1. Limited model scope
   Experiments in §3.2 only evaluate Qwen3-8B, a hybrid reasoning model with explicit Thinking / No-Thinking modes. It remains unclear whether the seeding mechanism generalizes to pure reasoning models that lack dual-mode design.

2. Unverified seed correctness
   Incorrect or biased seeds may negatively affect reasoning, yet §3.2 provides no quantitative analysis.

3. Single-seed assumption
   Each problem uses only one seed or many seed thoughts? If many, §3.2 does not explore multi-seed generation or ensemble strategies that could mitigate bad seeds.
---

§4.2 Difficulty Estimator
1. No reported accuracy
   §4.2 introduces a RoBERTa-based difficulty estimator but provides no quantitative results (accuracy, F1, etc.), making its reliability unclear.
---

§4.4 Speculative Decoding
1. No seed length statistics
   The paper does not report how many tokens each seed contains or their ratio within the total reasoning context.

2. Unclear need for speculative decoding
   - Seeds are short (≈50–150 tokens by estimation), so it is uncertain why speculative decoding is necessary for acceleration.
   - Since the seed tokens are already generated before reasoning begins, a standard prefill could directly process the full prompt (question + seed).  It remains unclear why speculative decoding is needed, what it specifically accelerates, and whether it actually participates in the later decoding process once the prefill is complete.
---

Figures and Tables
1. Figure 4 — unclear visualization
   Figure 4 is not clearly presented, with inconsistent colors and minor formatting issues, like typo "selfThink".

2. Table 1 — missing baseline without speculative decoding
   Table 1 lacks a SeedThink (without speculative decoding) variant.  Since other baselines use a single-model setup, adding speculative decoding introduces extra memory and compute usage, reducing fairness.

3. Table 1 — limited model diversity
   Table 1 only evaluates Qwen3-8B, without comparisons to more pure reasoning models (e.g., Phi-4). It is unclear whether improvements depend on the dual-mode (Think / No-Think) architecture or can generalize to standard reasoning-only models.

**Questions:**

Questions are in the weakness.

---

> ### Author Response · Authors · 2025-11-28
> **Response to Reviewer 7GgU [1/6]**
>
> We sincerely thank the reviewer for the time and effort spent reviewing our paper and for acknowledging the clarity and novelty of our framework, the performance gains, and the effective use of seeds. Below, we address the raised questions and concerns in detail:
>
> > **W1:** Limited model scope in §3.2 Observation
> > Experiments in §3.2 only evaluate Qwen3-8B, a hybrid reasoning model with explicit Thinking / No-Thinking modes. It remains unclear whether the seeding mechanism generalizes to pure reasoning models that lack dual-mode design.
>
> **A1:** To address the concern about model scope, we have conducted new experiments for Section 3.2 on **DeepSeek-R1-Distill-LLaMA-8B**, a model that fits the reviewer's description of a "pure reasoning model that lacks a dual-mode design." The new results, which we will include in the revised manuscript, confirm that the observed relationship between problem difficulty and optimal seed granularity holds for this model as well. This demonstrates that our finding is a **general phenomenon** and not specific to the Qwen3 architecture.
>
> ---
> > **W2:** Unverified seed correctness
> > Incorrect or biased seeds may negatively affect reasoning, yet §3.2 provides no quantitative analysis.
>
> ---
>
> **A2:** We conducted four separate runs of **SeedThink** on the Math500 dataset, each using a distinct seed prompt. Each run produced **500 seeds**, yielding **a total of 4×500 seeds**. We categorized each seed based on its content and its potential to introduce bias:
>
> - **Already correct:** The seed itself contains the final correct answer, making the subsequent reasoning a process of formal verification.
> - **No biased info:** The seed provides a neutral, high-level plan or structure without containing specific numerical values or assumptions that could be incorrect. This is the ideal "planning" scenario.
> - **Introduce biased info but not affect:** The seed contains a minor error or a questionable assumption, but the LRM, during its reasoning phase, identifies and corrects this flaw, arriving at the correct final answer. This demonstrates the model's robustness.
> - **Introduce biased info and affect:** The seed contains errors that mislead the LRM, causing the final answer to be incorrect. This represents the "catastrophic failure" scenario the reviewer is concerned about.
>
> **Key Findings from this analysis:**
>
> 1. **Cases of misguidance are infrequent.** The rightmost column shows that the number of instances where a flawed seed leads to an incorrect final answer is low, peaking at 7.0% for the most detailed seed types. Furthermore, we should also clarify that the above result comes from using a single seed type across all problem difficulties. In practice, our SeedThink strategy is adaptive: we only use detailed_step seeds for simple problems, where they are less likely to mislead the model.
> 2. **The results demonstrate that** **SeedThink** **possesses a self-correction capability**. The third column ("Biased Info, No Affect") shows that the model is not a naive follower. For example, when presented with 90 flawed seeds from the general_approach , it still adjusted its reasoning path to produce a correct final answer in 64 of those cases (**a 71% success rate**). This capability stems from a combination of the model's inherent ability to identify biased information and a reasoning stage prompt specifically designed to compel it to verify and correct the seed's contents.
>
> The results of this experiment is presented below:
>
> **Table 1: Analysis of various seed prompts and their effect on final answers.**
>
> |    **Seed Prompt**    | **Already Correct** | **No Biased Info** | **Biased Info, No Affect** | **Biased Info, Affects Final Answer** |
> | :-------------------: | :-----------------: | :----------------: | :------------------------: | :-----------------------------------: |
> |     outline seed      |          0          |   100% (500/500)   |             0              |                   0                   |
> |    key_steps seed     |   11.2% (56/500)    |  75.2% (376/500)   |       8.0% (40/500)        |             5.6% (28/500)             |
> | general_approach seed |   50.8% (254/500)   |  31.2% (156/500)   |       12.8% (64/500)       |             5.2% (26/500)             |
> |  detailed_steps seed  |   81.2%(406/500)    |   3.4% (17/500)    |       8.4% (42/500)        |             7.0% (35/500)             |

---

> ### Author Response · Authors · 2025-11-28
> **Response to Reviewer 7GgU [2/6]**
>
> > **W4:** No reported accuracy in §4.2 Difficulty Estimator
> > §4.2 introduces a RoBERTa-based difficulty estimator but provides no quantitative results (accuracy, F1, etc.), making its reliability unclear.
>
> **A3:** To address this, we will supplement **Appendix D** of the manuscript with **additional training details**. Furthermore, we present a comprehensive breakdown of the estimator's performance on the Math500 dataset. We go beyond simple classification accuracy to analyze the more important downstream effects: how do the estimator's predictions—both correct and incorrect—impact the final answer's accuracy and generation length?
>
> The analysis below shows that even when the estimator misclassifies a problem, it does not lead to a catastrophic failure. Instead, it results in a predictable and often acceptable trade-off between computational cost and solution accuracy. As the table shows, we can analyze the two distinct types of misclassification scenarios.
>
> The first type occurs when a **harder problem is misclassified as easier** (e.g., a True 'Hard' problem predicted as 'Medium'). In this case, the model receives a seed that is more detailed than optimal. While this can lead to a decrease in accuracy—for instance, 'Hard' problems misclassified as 'Medium' see their accuracy drop from 92.5% to 84.6%—this is coupled with a **corresponding reduction in generation length**.. The average generation length for these same problems plummets from **5258 to 3705 tokens**. This outcome represents a clear trade-off: a modest reduction in accuracy for a substantial improvement in computational speed.
>
> The second type of error is when an **easier problem is misclassified as more difficult** (e.g., a True 'Easy' problem predicted as 'Medium'). Here, the model is given a higher-level, more abstract seed than necessary. This prompts a more cautious reasoning process, which our data shows **actually increases accuracy**. All those misclassified types achieve **100% accuracy**, an improvement over the 96.8% for those classified correctly. This gain in robustness comes at the expected cost of efficiency, with the average token length increasing from 1206 to 1676. This scenario again illustrates a trade-off: sacrificing efficiency for a gain in accuracy.
>
> In summary, The vast majority of samples (422/500, or 84.4%) are classified correctly, achieving a high accuracy of **96.2%**. On the 78 misclassified instances, performance remains strong at **89.7% accuracy.**
>
> **Table 2. Confusion Matrix (Counts).**
>
> This table shows the number of items for each label combination of true and predicted labels.
>
> | True Label /Predicted Label |      Easy       |     Medium      |      Hard       |
> | :-----------------------------: | :-------------: | :-------------: | :-------------: |
> |            **Easy**             | **125** (94.0%) |        7        |        1        |
> |           **Medium**            |       30        | **191** (82.0%) |       12        |
> |            **Hard**             |        2        |       26        | **106** (79.1%) |
>
>
>
>
> **Table 3. Average Accuracy.**
>
> This table shows the average accuracy of the answers for the items in each cell.
>
> | True Label /Predicted Label |  Easy  | Medium |  Hard   |
> | :-----------------------------: | :----: | :----: | :-----: |
> |            **Easy**             | 96.80% |  100%  | 100.00% |
> |           **Medium**            | 90.00% | 97.91% | 100.00% |
> |            **Hard**             | 50.00% | 84.62% | 92.45%  |
>
>
>
>
> **Table 4. Average Length (in tokens).**
>
> This table shows the average length (in tokens) of the generated answers for the items in each cell.
>
> | True Label /Predicted Label |         Easy          |        Medium        |         Hard         |
> | :-----------------------------: | :-------------------: | :------------------: | :------------------: |
> |            **Easy**             |   515 + 691 = 1,206   | 411 + 1,265 = 1,676  | 245 + 2,608 =  2,853 |
> |           **Medium**            |  903 + 1,047 = 1,950  | 612 + 1,455 = 2,067  | 230 + 1,992 = 2,222  |
> |            **Hard**             | 1,417 + 1,688 = 3,105 | 1054 + 2,651 = 3,705 | 242 + 5,015  = 5,257 |

---

> ### Author Response · Authors · 2025-11-28
> **Response to Reviewer 7GgU [3/6]**
>
> > **W5:** No seed length statistics
> > The paper does not report how many tokens each seed contains or their ratio within the total reasoning context.
> > **W6-1:** Unclear need for speculative decoding
> > Seeds are short (≈50–150 tokens by estimation), so it is uncertain why speculative decoding is necessary for acceleration.
>
> ---
>
> **A4 & A5:** Below are the  generated seed length of different seed prompt.
>
> **Table5. Different seed prompt and it's generated seed length of Qwen3-14B on Math500 dataset**
>
> |         Seeds         | level-1 | level-2 | level-3 | level-4 | level-5 |
> | :-------------------: | :-----: | :-----: | :-----: | :-----: | :-----: |
> |     outline seed      |   137   |   150   |   179   |   203   |   243   |
> |    key_steps seed     |   274   |   333   |   415   |   514   |   711   |
> | general_approach seed |   302   |   394   |   521   |   912   |  1,391  |
> |  detailed_steps seed  |   585   |   685   |   862   |  1,110  |  1,461  |
>
>
> > **W6-2:** Unclear need for speculative decoding
> >
> > Since the seed tokens are already generated before reasoning begins, a standard prefill could directly process the full prompt (question + seed). It remains unclear why speculative decoding is needed, what it specifically accelerates, and whether it actually participates in the later decoding process once the prefill is complete.
>
> ---
>
> We apologize for not detailing this more clearly in our manuscript.
>
> **1. Why is speculative decoding needed?**
>
> While the seed reduces reasoning length by guiding the model toward a shorter and more direct reasoning path, this improvement only reduces the **number** of tokens generated. Speculative decoding (SD) offers a **complementary benefit** by accelerating the **speed** of token generation.
>
> **2. What does it specifically accelerate?**
>
> SD accelerates the **Think-stage decoding.** This is where the majority of the runtime is spent, so improvements here materially reduce wall-clock latency.
>
> **3. Whether it actually participates in the later decoding process once the prefill is complete?**
>
> Yes. SD is applied **after** the full prompt (question + seed) is prefixed and the model finishes encoding the context. Prefill only loads the prompt; SD continues to operate for **every subsequent generation step** during the Think stage.
>
> By enhancing model-free, suffix matching based speculative decoding methods (existing work like SuffixDecoding [1], SAM decoding [2]). We repurpose the generated "seed"  as a  **higher-quality source for predicting and drafting future tokens to warmup speculative decoding**, which improves draft quality and accept rate without modifying the draft and verification logic. This approach is inherently low-overhead and training-free. **Such a training-free warmup for speculative decoding using generated tokens in the fast thinking mode is novel and not studied in the previous literature.**
>
> **In summary**, our contribution is not the invention of model-free, suffix-matching-based speculative decoding, but the insight that our SeedThink framework **creates ideal conditions for it to thrive.** The seed itself serves as a powerful, on-the-fly source for speculative drafts.
>
> To empirically validate our claim that SeedThink naturally enhances suffix matching based speculative decoding, we conducted a detailed ablation study reported in table below. We find that **longer seeds generally increase speculative gains** (speed up from **1.47 to 1.64** and MAT from **1.65 to 1.96**), as they produce more opportunities for suffix matches.
>
> **Table 6.  Speculative gains (using SAM decoding) of different length seeds on the Math500 dataset, MAT short for Mean Accept Token.**
>
> |              Seeds               | MAT  | Throughput(tokens/s) | Speed up from SD | Length | Accuracy |
> | :------------------------------: | :--: | :------------------: | :--------------: | :----: | :------: |
> |          Think(w/o SAM)          |  0   |        28.44         |        -         |  4741  |  96.8%   |
> |          Think(w/ SAM)           | 1.65 |        40.26         |      × 1.42      |  4776  |  96.8%   |
> |     SeedThink(outline seed)      | 1.73 |        41.72         |      × 1.47      |  2959  |  93.8%   |
> |    SeedThink(key_steps seed)     | 1.82 |        43.88         |      × 1.54      |  2847  |  93.4%   |
> | SeedThink(general_approach seed) | 1.88 |        45.04         |      × 1.58      |  2735  |  95.6%   |
> |  SeedThink(detailed_steps seed)  | 1.96 |        46.87         |      × 1.64      |  2650  |  94.4%   |
>
> [1] Gabriele Oliaro, Zhihao Jia, Daniel F Campos, and Aurick Qiao. Suffixdecoding: Extreme speculative decoding for emerging ai applications. In The Thirty-ninth Annual Conference on Neural Information Processing Systems, 2025.
>
> [2] Yuxuan Hu, Ke Wang, Xiaokang Zhang, Fanjin Zhang, Cuiping Li, Hong Chen, and Jing Zhang.Sam decoding: Speculative decoding via suffix automaton. arXiv preprint arXiv:2411.10666,2024.

---

> ### Author Response · Authors · 2025-11-28
> **Response to Reviewer 7GgU [4/6]**
>
> >  **W3:** Single-seed assumption
> >  Each problem uses only one seed or many seed thoughts? If many, §3.2 does not explore multi-seed generation or ensemble strategies that could mitigate bad seeds.
>
> ---
>
> **A6:** In our current SeedThink framework, we generate a single seed thought for each problem.  While our experiments show high accuracy, suggesting our seeds are generally effective, we have carefully considered the multi-seed alternative. Our decision to use a single seed is based on a deliberate focus on optimizing for **single-pass output length and throughput**, where the overhead of generating and managing multiple seeds becomes a critical factor.
>
> Here are the key considerations:
>
> 1. **The variable cost of the seed:** The seed is not a fixed-size entity. As our analysis shows, its length and complexity vary dynamically based on the seed prompt and problem difficulty. For easy problems, a detailed_steps seed comprises **30-40%** of the output tokens, so a multi-seed approach negates its benefits. For difficult problems, the limited gains from a short outline seed mean that a multi-seed strategy still conflicts with our efficiency-oriented design.
>
> 2. **The overhead of multi-seed selection/ensembling:** A multi-seed approach necessitates an additional mechanism for selection or ensembling, which introduces its own complexity and latency:
>
>    - **Screening:** How do we select the "best" seed without running the full, expensive reasoning for each one? This would require a fast, reliable verifier, adding another layer of complexity.
>
>    - **Voting/Ensembling:** Running the full reasoning for all seeds and taking a majority vote is a form of ensembling. This is a valid strategy to maximize accuracy, but it fundamentally shifts the goal away from single-pass efficiency.
>
> 3. **A different parallel thinking paradigm:** We agree with the reviewer that a multi-seed strategy is an excellent idea, but we believe it is better suited for a different paradigm, such as improving **accuracy** in scenarios like **parallel thinking,** which often use **pass@k** for evaluation. In that context, generating multiple parallel reasoning paths is the intended mechanism. Our work, however, is focused on accelerating the single-pass efficiency.
>
> In conclusion, while the concept of multi-seed generation is powerful for maximizing accuracy via parallel exploration, it introduces overheads that are counterproductive to our core goal of single-pass efficiency.
>
>
> > **W7** Figure 4 — unclear visualization
> > Figure 4 is not clearly presented, with inconsistent colors and minor formatting issues, like typo "selfThink".
>
> **A7:** We thank the reviewer for pointing out the visualization issues in Figure 4. We will revise the figure in our manuscript to ensure consistent color usage and uniform styling across all subplots. We also corrected the typo from "selfThink" to "SeedThink."

---

> ### Author Response · Authors · 2025-11-28
> **Response to Reviewer 7GgU [5/6]**
>
> > **W8:** Table 1 — missing baseline without speculative decoding
> > Table 1 lacks a SeedThink (without SD) variant. Since other baselines use a single-model setup, adding speculative decoding introduces extra memory and compute usage, reducing fairness.
>
> ---
>
> **A8:** Although we do not include a SeedThink variant (w/o SD) in our manuscript (Table 1) , kindly note that speculative decoding by design does **not** alter the final output token sequence; it merely accelerates inference through a draft-then-verify procedure. Therefore, the length and accuracy metrics we report reflect exactly the same-quality output.
>
> Moreover, we include a dedicated ablation study (Section 5.3) which directly measures the impact of SD on inference speed, thereby empirically demonstrating the efficiency gains of our dual-path design. Based on this, we believe our current setup suffices to show the contribution of speculative decoding to inference speed clearly and fairly.
>
> Regarding the reviewer's concern about fairness, we conducted new experiments **disabled speculative decoding to ensure a fair comparison.** This isolates the throughput gains that result purely from SeedThink's ability to reduce the total number of generated tokens. The experiments evaluating end-to-end system performance in a multi-batch setting. Using vLLM, a state-of-the-art serving engine, we benchmarked performance on the MATH500 dataset with four models: the GPT-OSS-20B model and three Qwen3 models. To simulate a realistic, high-load scenario, we set the maximum vLLM batch size to 256 for all methods, which is well below the 500 total requests.
>
> The results, presented below, report the **total runtime** required to process the entire 500-sample dataset, a direct measure of system throughput. From these results, we conclude that **SeedThink reduces total runtime by 24.7% to 43.2% with only a marginal 0.2% to 2.0% drop** in accuracy. This performance surpasses the CoThink baseline in runtime reduction while achieving comparable accuracy across all four models.
>
>
> **Table 7. Result of GPT-OSS-20B.**
>
> |    Methods    | Total Runtime (s) | Average Runtime (s) | Average Tokens | Accuracy |
> | :-----------: | :---------------: | :-----------------: | :------------: | :------: |
> |     Think     |       840.5       |        134.4        |     4,380      |  97.0%   |
> |    NoThink    |       52.7        |        14.8         |      446       |  89.2%   |
> |    CoThink    |       721.5       |        123.0        |     3,867      |  95.0%   |
> | **SeedThink** |       633.3       |        104.7        |     3,325      |  95.0%   |
>
> **Table 8. Result of Qwen3-8B.**
>
> |    Methods    | Total Runtime (s) | Average Runtime (s) | Average Tokens | Accuracy |
> | :-----------: | :---------------: | :-----------------: | :------------: | :------: |
> |     Think     |      1,441.3      |        367.2        |     5,033      |  97.2%   |
> |    NoThink    |       436.8       |        41.2         |      827       |  85.4%   |
> |    CoThink    |      1,172.0      |        303.2        |     4,044      |  97.0%   |
> | **SeedThink** |      1,019.3      |        252.7        |     3,483      |  96.8%   |
>
> **Table 9. Result of Qwen3-14B.**
>
> |    Methods    | Total Runtime (s) | Average Runtime (s) | Average Tokens | Accuracy |
> | :-----------: | :---------------: | :-----------------: | :------------: | :------: |
> |     Think     |      1,511.9      |        385.2        |     4,477      |  97.4%   |
> |    NoThink    |       165.2       |        47.1         |      677       |  86.0%   |
> |    CoThink    |      1,355.4      |        306.0        |     3,526      |  97.0%   |
> | **SeedThink** |      1,064.7      |        230.5        |     2,730      |  97.2%   |
>
> **Table 10. Result of Qwen3-32B.**
>
> |    Methods    | Total Runtime (s) | Average Runtime (s) | Average Tokens | Accuracy |
> | :-----------: | :---------------: | :-----------------: | :------------: | :------: |
> |     Think     |      3,244.3      |        873.9        |     4,299      |  97.6%   |
> |    NoThink    |      1,120.0      |        75.8         |      742       |  88.0%   |
> |    CoThink    |      2,314.6      |        578.1        |     3,289      |  96.8%   |
> | **SeedThink** |      1,842.9      |        362.6        |     2,512      |  96.6%   |

---

> ### Author Response · Authors · 2025-11-28
> **Response to Reviewer 7GgU [6/6]**
>
> > **W9:** Table 1 — limited model diversity
> >
> > Table 1 only evaluates Qwen3-8B, without comparisons to more pure reasoning models (e.g., Phi-4). It is unclear whether improvements depend on the dual-mode (Think / No-Think) architecture or can generalize to standard reasoning-only models.
>
> **A9:** We would like to clarify a potential misunderstanding regarding the experiments presented in Table 1. The reviewer noted that the table "only evaluates Qwen3-8B," but our experiments in that table actually cover a more diverse set of models: **Qwen3-8B, Qwen3-14B, Qwen3-32B, and DeepSeek-R1-Distill-LLaMA-8B**.  In addition, we further validate the generalization of the proposed meta thinking framework on **GPT-OSS-20B**, the high-end, open-source, MoE model recently released by OpenAI. In which, SeedThink outperforms the strong baseline CoThink by 12.2% in terms of practical multi-batch runtime reduction using vLLM, with the same accuracy performance.
>
> Crucially, our evaluation includes **DeepSeek-R1-Distill-LLaMA-8B**. This model is built upon the standard LLaMA decoder-only architecture and is well-regarded for its reasoning capabilities. It is precisely the type of "pure reasoning model" the reviewer requested, as it does not have a specialized "dual-mode" architecture. Our method achieved a **2.3x speedup** on this model while maintaining competitive accuracy, confirming that SeedThink is not dependent on dual-mode (Think / No-Think) architecture and generalizes effectively to standard reasoning-only LRMs.

---

### Official Review · Reviewer_mFyz · 2025-10-31

**Soundness:** 3
**Presentation:** 3
**Contribution:** 2
**Rating:** 2
**Confidence:** 4

**Summary:**

This paper focuses on the inefficiency of Large Reasoning Models (LRMs), which often exhibit "overthinking" behaviors like repetitive verification during complex chain-of-thought (CoT) reasoning. The authors propose SeedThink, a framework designed to improve efficiency by initializing the LRM's reasoning with a "seed thought." The core idea is based on the finding that the optimal granularity of this seed is not fixed, but depends on the problem's difficulty. Consequently, the SeedThink framework operates in a multi-stage process. First, a difficulty classifier (a RoBERTa model) estimates the problem's complexity. Second, based on this classification, a seed thought of adaptive granularity (ranging from a high-level outline for hard problems to detailed steps for easy ones) is generated by the LRM itself using specific prompts. Finally, this seed is used to achieve a "dual-path acceleration": it not only guides the LRM to produce shorter, more efficient reasoning traces (by pruning unproductive branches) but is also repurposed as a high-quality, model-free draft corpus to enhance speculative decoding, thereby accelerating token generation speed. The authors conduct experiments, primarily on math benchmarks like GSM8K and MATH, to demonstrate that their method can achieve significant speedups  while maintaining competitive accuracy.

**Strengths:**

1. The systematic study in Section 3.2 demonstrates a non-monotonic relationship between optimal seed-thought granularity and problem difficulty (i.e., detailed seeds for easy problems, high-level outlines for hard problems). This is a useful insight for future work on adaptive reasoning.
2. The method achieves significant speedups (reported up to 4.1x) while maintaining competitive accuracy on the tested math benchmarks.

**Weaknesses:**

1. The proposed method is a complex, sequential pipeline (Classifier -> Seed Generation -> Guided Reasoning). This multi-stage approach is inherently fragile: an error in an early stage (e.g., the classifier misjudging a "hard" problem as "easy") will cascade and force a suboptimal strategy, potentially harming both accuracy and efficiency.
2. The paper focuses exclusively on single-instance latency (speedup per sample). In real-world serving scenarios, the critical metric is often system throughput (e.g., total tokens/sec). A multi-stage pipeline introduces significant scheduling, data handling, and I/O overhead that can severely degrade overall system throughput, even if individual sample latency is reduced. The paper does not analyze this trade-off.
3. The framework's reliance on a separate RoBERTa classifier is a significant practical weakness. This adds non-trivial overhead for training, deployment, and maintenance. More importantly, the classifier's stability and generalization are critical points of failure. The classifier was only tested on a single math dataset, and its ability to generalize to other domains (e.g., coding, general QA) is unproven.
4. The seed generation relies on a set of pre-defined, manually-crafted prompts (Appendix C). This approach is brittle, contains strong human priors, and is unlikely to generalize to new tasks or domains without significant, task-specific prompt engineering.
5. The evaluation is heavily concentrated on mathematical reasoning datasets (MATH, GSM8K) and primarily on the Qwen3 model family.

**Questions:**

Could the authors provide a more detailed explanation of how the speculative decoding mechanism is implemented? The paper states the seed is used as a "model-free" corpus. Is the seed simply used as a static prefix?

---

> ### Author Response · Authors · 2025-11-28
> **Response to Reviewer mFyz [1/5]**
>
> We sincerely thank the reviewer for the time and effort spent reviewing our paper and for acknowledging both the novel insights of our study and the efficiency gains our method provides. Below, we address the raised questions and concerns in detail:
>
> >  **W1:** The proposed method is a complex, sequential pipeline (Classifier -> Seed Generation -> Guided Reasoning). This multi-stage approach is inherently fragile: an error in an early stage (e.g., the classifier misjudging a "hard" problem as "easy") will cascade and force a suboptimal strategy, potentially harming both accuracy and efficiency.
>
> **A1:** We thank the insightful comment regarding the potential fragility of our sequential pipeline. To address this, we analyzed our  pipeline's behavior under misclassification on the Math500 dataset. We present the results in tables 1,2,3 below, which detail not only the classification counts but also the resulting average accuracy and token length for each true-vs-predicted category (based on Qwen3-32B).
>
> As the tables below show, we can analyze the two distinct types of misclassification scenarios.
>
> - The first type occurs when a **harder problem is misclassified as easier** (e.g., the cell for True 'Hard' / Predicted 'Medium'). In this case, the model receives a seed that is more detailed than optimal. While this can lead to a decrease in accuracy—for instance, 'Hard' problems misclassified as 'Medium' see their accuracy drop from 92.5% to 84.6%—this is coupled with a **corresponding reduction in generation length.** The average generation length for these same problems decreases from **5258 to 3705 tokens**.
> - The second type of error is when an **easier problem is misclassified as more difficult** (e.g., the cell for True 'Easy' / Predicted 'Medium'). Here, the model is given a higher-level, more abstract seed than necessary. This prompts a more cautious reasoning process, which our data shows **actually increases accuracy**. All those misclassified types achieve **100% accuracy**, an improvement over the 96.8% for those classified correctly. This gain in robustness comes at the expected cost of efficiency, with the average token length increasing from 1206 to 1676.
>
> In summary, the fear of a cascaded failure that harms both key metrics is not borne out by the evidence. This data demonstrates that the pipeline is not fragile in the manner feared; instead, classification errors lead to a **predictable trade-off**, and crucially, we found no case where both metrics are simultaneously degraded. The vast majority of samples (422/500, or 84.4%) are classified correctly, achieving a high accuracy of **96.2%**. On the 78 misclassified instances, performance remains strong at **89.7% accuracy**.
>
> **Table 1. Confusion Matrix (Counts).**
>
> This table shows the number of items for each label combination of true and predicted labels.
>
> | True Label /Predicted Label |      Easy       |     Medium      |      Hard       |
> | :-----------------------------: | :-------------: | :-------------: | :-------------: |
> |            **Easy**             | **125** (94.0%) |        7        |        1        |
> |           **Medium**            |       30        | **191** (82.0%) |       12        |
> |            **Hard**             |        2        |       26        | **106** (79.1%) |
>
> **Table 2. Average Accuracy.**
>
> This table shows the average accuracy of the answers for the items in each cell.
>
> | True Label /Predicted Label |  Easy  | Medium |  Hard   |
> | :-----------------------------: | :----: | :----: | :-----: |
> |            **Easy**             | 96.80% |  100%  | 100.00% |
> |           **Medium**            | 90.00% | 97.91% | 100.00% |
> |            **Hard**             | 50.00% | 84.62% | 92.45%  |
>
> **Table 3. Average Length (in tokens).**
>
> This table shows the average length (in tokens) of the generated answers for the items in each cell.
>
> | True Label /Predicted Label |         Easy          |        Medium        |         Hard         |
> | :-----------------------------: | :-------------------: | :------------------: | :------------------: |
> |            **Easy**             |   515 + 691 = 1,206   | 411 + 1,265 = 1,676  | 245 + 2,608 =  2,853 |
> |           **Medium**            |  903 + 1,047 = 1,950  | 612 + 1,455 = 2,067  | 230 + 1,992 = 2,222  |
> |            **Hard**             | 1,417 + 1,688 = 3,105 | 1054 + 2,651 = 3,705 | 242 + 5,015  = 5,257 |

---

> ### Author Response · Authors · 2025-11-28
> **Response to Reviewer mFyz [2/5]**
>
> > **W2:** The paper focuses exclusively on single-instance latency (speedup per sample). In real-world serving scenarios, the critical metric is often system throughput (e.g., total tokens/sec). A multi-stage pipeline introduces significant scheduling, data handling, and I/O overhead that can severely degrade overall system throughput, even if individual sample latency is reduced. The paper does not analyze this trade-off.
>
> **A2:** To empirically investigate this trade-off, we conducted new experiments evaluating end-to-end system performance in a multi-batch setting. Using vLLM, a state-of-the-art serving engine, we benchmarked performance on the MATH500 dataset with four models: the GPT-OSS-20B model and three Qwen3 models.
>
> The results, presented below, report the **total runtime** required to process the entire 500-sample dataset, a direct measure of system throughput. From these results, we conclude that **SeedThink reduces total runtime by 24.7% to 43.2% with only a marginal 0.2% to 2.0% drop** in accuracy. This performance surpasses the CoThink baseline in runtime reduction while achieving comparable accuracy across all four models.
>
> For this analysis, we disabled speculative decoding to ensure a fair comparison. This isolates the throughput gains that result purely from SeedThink's ability to reduce the total number of generated tokens. To simulate a realistic, high-load scenario, we set the maximum vLLM batch size to 256 for all methods, which is well below the 500 total requests. This setup validates SeedThink's effectiveness for system-level speedup by accounting for practical multi-batch scheduling across its entire three-stage pipeline.
>
>
> **Table 4. Result of GPT-OSS-20B.**
>
> |    Methods    | Total Runtime (s) | Average Runtime (s) | Average Tokens | Accuracy |
> | :-----------: | :---------------: | :-----------------: | :------------: | :------: |
> |     Think     |       840.5       |        134.4        |     4,380      |  97.0%   |
> |    NoThink    |       52.7        |        14.8         |      446       |  89.2%   |
> |    CoThink    |       721.5       |        123.0        |     3,867      |  95.0%   |
> | **SeedThink** |       633.3       |        104.7        |     3,325      |  95.0%   |
>
> **Table 5. Result of Qwen3-8B.**
>
> |    Methods    | Total Runtime (s) | Average Runtime (s) | Average Tokens | Accuracy |
> | :-----------: | :---------------: | :-----------------: | :------------: | :------: |
> |     Think     |      1,441.3      |        367.2        |     5,033      |  97.2%   |
> |    NoThink    |       436.8       |        41.2         |      827       |  85.4%   |
> |    CoThink    |      1,172.0      |        303.2        |     4,044      |  97.0%   |
> | **SeedThink** |      1,019.3      |        252.7        |     3,483      |  96.8%   |
>
> **Table 6. Result of Qwen3-14B.**
>
> |    Methods    | Total Runtime (s) | Average Runtime (s) | Average Tokens | Accuracy |
> | :-----------: | :---------------: | :-----------------: | :------------: | :------: |
> |     Think     |      1,511.9      |        385.2        |     4,477      |  97.4%   |
> |    NoThink    |       165.2       |        47.1         |      677       |  86.0%   |
> |    CoThink    |      1,355.4      |        306.0        |     3,526      |  97.0%   |
> | **SeedThink** |      1,064.7      |        230.5        |     2,730      |  97.2%   |
>
> **Table 7. Result of Qwen3-32B.**
>
> |    Methods    | Total Runtime (s) | Average Runtime (s) | Average Tokens | Accuracy |
> | :-----------: | :---------------: | :-----------------: | :------------: | :------: |
> |     Think     |      3,244.3      |        873.9        |     4,299      |  97.6%   |
> |    NoThink    |      1,120.0      |        75.8         |      742       |  88.0%   |
> |    CoThink    |      2,314.6      |        578.1        |     3,289      |  96.8%   |
> | **SeedThink** |      1,842.9      |        362.6        |     2,512      |  96.6%   |

---

> ### Author Response · Authors · 2025-11-28
> **Response to Reviewer mFyz [3/5]**
>
> > **W3:** The framework's reliance on a separate RoBERTa classifier is a significant practical weakness. This adds non-trivial overhead for training, deployment, and maintenance. More importantly, the classifier's stability and generalization are critical points of failure. The classifier was only tested on a single math dataset, and its ability to generalize to other domains (e.g., coding, general QA) is unproven.
>
> **A3:** We admit that practicality and generalization of the classifier are indeed important considerations. We would like to address these points as follows:
>
> 1. **The training and deployment overhead is minimal.** We want to clarify that the overhead for the RoBERTa model as a difficulty estimator is not a significant practical weakness. Specifically, the RoBERTa-base is a compact **125M** parameter model that can be trained in under 60 minutes on a single RTX 4060 GPU. It is also highly efficient, allowing for deployment on a single **CPU** with an inference latency of less than 1 second per query.
> 2. **Our framework is modular by design.** We chose a lightweight RoBERTa model specifically for extreme efficiency, ensuring that the difficulty estimator adds minimal latency. However, our overall framework is not dependent on this specific classifier. The difficulty estimator is a modular component, and it can be replaced with any other model (or even an entirely different mechanism) that can classify problem complexity. This design provides flexibility and allows practitioners to use more sophisticated classifiers if their specific domain or performance requirements call for it.
> 3. **This approach aligns with established industry practices.** Using a fine-tuned small model for semantic routing is a mature and effective solution in production environments. A prominent example is the vllm-semantic-router [1], a popular open-source library that also relies on fine-tuning a small BERT-style model to route queries. This demonstrates that our choice is a practical and widely used approach.
>
> [1] https://github.com/vllm-project/semantic-router

---

> ### Author Response · Authors · 2025-11-28
> **Response to Reviewer mFyz [4/5]**
>
> > **W4:** The seed generation relies on a set of pre-defined, manually-crafted prompts (Appendix C). This approach is brittle, contains strong human priors, and is unlikely to generalize to new tasks or domains without significant, task-specific prompt engineering.
> >
> > **W5:** The evaluation is heavily concentrated on mathematical reasoning datasets (MATH, GSM8K) and primarily on the Qwen3 model family.
>
> **A4 & A5:** To address the valid concern (**W5-1**) that our evaluation was heavily concentrated on mathematical reasoning datasets, we have conducted new experiments use Qwen3-32B on two challenging, non-mathematical reasoning datasets: **GPQA-Diamond** (complex scientific questions for graduate-level experts) and **FinEval** (expert-level benchmark for finance domain knowledge). These datasets were chosen specifically to test SeedThink's effectiveness in diverse, expert-level domains **without any further prompt-tuning**. Regarding the concern (**W5-2**)  that our evaluation focuses primarily on the Qwen3 model family, we also tested the **new GPT-OSS-20B model**, with the results reported in **A2**.
>
>   These datasets do not provide the difficulty labels required to train or validate our difficulty estimator. Therefore, for these experiments, we used a fixed seed prompt (detailed_steps) for all problems. This setup directly tests the reviewer's concern (**W4**) that our method requires significant task-specific prompt engineering to generalize.
>
>   These results, listed in the tables below, lead to two important conclusions:
>
>   - **Generalization to new domains:** SeedThink consistently outperforms NoThink and CoThink and achieves the closest accuracy to the Think mode baseline, while persistently delivering a **30-35% token reduction** on both the finance and science datasets. This demonstrates that the core benefit of our framework—maintaining high performance with significantly improved efficiency—is generalizable beyond mathematics.
>   - **Robustness of seed prompts:** These results were achieved using the same pre-defined seed prompt we used in mathematical reasoning datasets without any domain-specific tuning. This directly contradicts the hypothesis that the approach is brittle and requires significant manual effort to adapt to new tasks.
>
> **Table 8. Performance on professional finance domain reasoning (FinEval).**
>
> |    Methods    | Average Tokens | Average Runtimes(s) | Accuracy |
> | :-----------: | -------------: | ------------------: | -------: |
> |     Think     |          2,606 |               157.0 |   68.64% |
> |    NoThink    |            498 |                29.5 |   64.29% |
> |    CoThink    |          1,926 |               110.7 |   56.30% |
> | **SeedThink** |          1,814 |                59.9 |   67.94% |
>
> **Table 9. Performance on complex scientific reasoning (GPQA-Diamond).**
>
> |    Methods    | Average Tokens | Average Runtimes(s) | Accuracy |
> | :-----------: | -------------: | ------------------: | -------: |
> |     Think     |          7,643 |               459.3 |   66.16% |
> |    NoThink    |          1,123 |                64.1 |   54.55% |
> |    CoThink    |          5,497 |               315.9 |   58.08% |
> | **SeedThink** |          5,013 |               137.6 |   62.63% |

---

> ### Author Response · Authors · 2025-11-28
> **Response to Reviewer mFyz [5/5]**
>
> > **Q1:** Could the authors provide a more detailed explanation of how the speculative decoding mechanism is implemented? The paper states the seed is used as a "model-free" corpus. Is the seed simply used as a static prefix?
>
> **A6:** We apologise for not detailing this more clearly in our manuscript.
>
> - To clarify, our implementation is based on the Spec_Bench [1], a widely used speculative decoding benchmark which provides implementations of existing speculative decoding methods. Our SeedThink framework is a natural and highly effective fit for a class of efficient, training-free speculative decoding methods, which is based on n-gram level suffix matching (existing work like SuffixDecoding [2], SAM decoding [3]). The exact method we use in SeedThink is SAM decoding, due to its superior performance in this category.
> - Regarding your question, "Is the seed simply used as a static prefix?": No, the seed is not just a static prefix; it serves as a high-quality, dynamic look-up corpus for predicting and drafting future tokens. We have not altered how SAM decoding works internally. Instead, **our key contribution is providing it with a higher-quality source for its speculative corpus**. By generating a semantically related "seed" of thought before the final answer, we ensure the suffix matching mechanism has access to more relevant and coherent text. **Such a training-free warmup for speculative decoding using generated tokens in the fast thinking mode leads to more accurate drafts and higher acceleration**.
> - Furthermore, our approach can be combined with training-based speculative decoding methods, dynamically switching between two sources of draft tokens. As shown in our manuscript, we integrated SeedThink with the SOTA training-based speculative method (Eagle3 [4]) and observed greater performance gains. This highlights the flexibility of our framework.
>
> [1] https://github.com/hemingkx/Spec-Bench
>
> [2] Gabriele Oliaro, Zhihao Jia, Daniel F Campos, and Aurick Qiao. Suffixdecoding: Extreme speculative decoding for emerging ai applications. In The Thirty-ninth Annual Conference on Neural Information Processing Systems, 2025.
>
> [3] Yuxuan Hu, Ke Wang, Xiaokang Zhang, Fanjin Zhang, Cuiping Li, Hong Chen, and Jing Zhang. Sam decoding: Speculative decoding via suffix automaton. arXiv preprint arXiv:2411.10666,2024.
>
> [4] Yuhui Li, Fangyun Wei, Chao Zhang, and Hongyang Zhang. Eagle-3: Scaling up inference acceleration of large language models via training-time test. arXiv preprint arXiv:2503.01840, 2025b.

---

### Official Review · Reviewer_2XUz · 2025-11-01

**Soundness:** 2
**Presentation:** 3
**Contribution:** 2
**Rating:** 6
**Confidence:** 3

**Summary:**

This paper addresses the inefficiency of Large Reasoning Models (LRMs) which, despite their strong performance, often suffer from "overthinking"—generating excessively long and redundant reasoning paths. The authors' core insight is that initializing the reasoning process with a "seed thought" can significantly shorten the reasoning trajectory. Critically, they observe and demonstrate through a pilot study that the optimal granularity of this seed (from a high-level outline to a detailed solution) is dependent on the problem's difficulty.

Based on this, the paper proposes SeedThink, a training-free framework that operates in two stages. First, it uses a difficulty estimator to assess the input problem's complexity. Second, it generates a seed thought with an appropriate level of detail (e.g., an outline for hard problems, detailed steps for easy ones). The final, extended reasoning is then generated conditioned on this seed. A key contribution is the dual use of this seed: not only does it guide the reasoning search to be more efficient, but its token sequence is also repurposed as a high-quality draft for speculative decoding, enabling faster token generation. The authors term this "dual-path acceleration". Experiments on math benchmarks (GSM8K, MATH500, AIME 2024) show that SeedThink achieves significant end-to-end speedups (up to 4.1×) and reduces generation length by up to 68%, with only a minimal drop in accuracy.

**Strengths:**

While prior work has explored using outlines to guide reasoning (e.g., CoThink), the idea of adapting the granularity of the initial guidance based on an estimate of problem difficulty is a novel and powerful concept. It moves beyond a one-size-fits-all approach to a more nuanced control strategy.

**Weaknesses:**

The evaluation is performed exclusively on mathematical reasoning tasks. While these are excellent benchmarks for complex reasoning, the paper's claims are about a general framework for LRMs. Its applicability and effectiveness on other reasoning-intensive domains, such as code generation, logical reasoning (e.g., LogiQA), or long-form question answering, remain unproven. Demonstrating the method's value on at least one non-math domain would significantly strengthen the paper's claims of generalizability.

**Questions:**

Please refer to Weaknesses.

---

> ### Author Response · Authors · 2025-11-28
> **Response to Reviewer 2XUz [1/1]**
>
> We sincerely thank the reviewer for the time and effort spent reviewing our paper. Below, we address the raised question and concern in detail:
>
> > **W1:** The evaluation is performed exclusively on mathematical reasoning tasks. While these are excellent benchmarks for complex reasoning, the paper's claims are about a general framework for LRMs. Its applicability and effectiveness on other reasoning-intensive domains, such as code generation, logical reasoning (e.g., LogiQA), or long-form question answering, remain unproven. Demonstrating the method's value on at least one non-math domain would significantly strengthen the paper's claims of generalizability.
>
> **A1:** We have conducted new experiments use Qwen3-32B on two challenging, non-mathematical reasoning datasets: **GPQA-Diamond** (complex scientific questions for graduate-level experts) and **FinEval** (expert-level benchmark for finance domain knowledge). These datasets were chosen specifically to test SeedThink's effectiveness in diverse, expert-level domains. Our key findings are listed below:
>
> - **Generalized to new domains**: SeedThink consistently outperforms NoThink and CoThink and achieves the closest accuracy to the Think mode, while persistently delivering a **30-35% token reduction** on both the finance and science datasets. This demonstrates that the core benefit of our framework—maintaining high performance with significantly improved efficiency—is generalizable beyond mathematics.
>
> - **Robustness of seed prompts:** These results were achieved using the same pre-defined seed prompt we used in mathematical reasoning datasets without any domain-specific tuning.
>
> **Table 1. Performance on professional finance domain reasoning (FinEval).**
>
> |    Methods    | Average Tokens | Average Runtimes(s) | Accuracy |
> | :-----------: | -------------: | ------------------: | -------: |
> |     Think     |          2,606 |               157.0 |   68.64% |
> |    NoThink    |            498 |                29.5 |   64.29% |
> |    CoThink    |          1,926 |               110.7 |   56.30% |
> | **SeedThink** |          1,814 |                59.9 |   67.94% |
>
> **Table 2. Performance on complex scientific reasoning (GPQA-Diamond).**
>
> |    Methods    | Average Tokens | Average Runtimes(s) | Accuracy |
> | :-----------: | -------------: | ------------------: | -------: |
> |     Think     |          7,643 |               459.3 |   66.16% |
> |    NoThink    |          1,123 |                64.1 |   54.55% |
> |    CoThink    |          5,497 |               315.9 |   58.08% |
> | **SeedThink** |          5,013 |               137.6 |   62.63% |

---

### Official Review · Reviewer_8k81 · 2025-11-02

**Soundness:** 2
**Presentation:** 3
**Contribution:** 2
**Rating:** 4
**Confidence:** 4

**Summary:**

SeedThink is a reasoning strategy that gives an initial thought called a “seed” that then sets the tone for all future reasoning tokens. This seed is computed based off a preliminary difficulty estimation, which quantifies how tough the problem is in terms of easy, moderate, and hard. Depending on what this difficulty is, the seed is then designed to reflect that complexity. Whatever follows from this seed must be deducible from it, making the seed what the authors call a “reasoning anchor”. The motivation is that large reasoning models (LRMs) struggle with overthinking in the form of too much verification or too many branching reasoning paths such as that found in a multi-path reasoning strategy like Tree of Thoughts or Graph of Thoughts. With this initial seed, the LRM now has a northern star to guide its thinking so that it might be able to avoid these traps.

**Strengths:**

*	The paper introduces adaptive seed initialization  to mitigate inefficiencies from “overthinking” in chain-of-thought models. It uses structured prior reasoning to prune unproductive reasoning paths.
*	Demonstrates up to 4.1× end-to-end acceleration and 68% reduction in token length with minimal accuracy degradation on multiple math benchmarks (GSM8K, MATH500, AIME 2024) and outperforms baselines like CoThink and JointThink in efficiency–accuracy trade-offs.
*	Creative reuse of seed thoughts as speculative drafts for acceleration.
*	Demonstration of how seed granularity interacts with task difficulty and reasoning length and includes ablation on speculative decoding and reflection/branching token reduction
*	The framework could potentially apply to any reasoning-heavy LLM, suggesting broader implications for efficient inference and adaptive reasoning.

**Weaknesses:**

•	The proposed “outline” seeding policy in SeedThink appears conceptually similar to the *Skeleton of Thought* technique presented at ICLR 2024. Although the authors briefly acknowledge this, the overlap in motivation and mechanism warrants deeper discussion.

•	The study omits direct comparisons with several key test-time reasoning or preemptive inference strategies such as *ERCogQA, SpecSearch, and Skeleton of Thought*. These represent state-of-the-art techniques that also focus on optimizing inference efficiency and reasoning control.

•	The practical efficiency of SeedThink may depend on the underlying LLM serving environment. For instance, differences in batch scheduling, token parallelism, or speculative decoding infrastructure could significantly affect latency and throughput.

•	The main novelty lies in combining existing paradigms (difficulty estimation, seeding, speculative decoding), not necessarily in new architectures or algorithms.

•	The difficulty estimator is a straightforward text classifier (RoBERTa), not jointly trained or self-adaptive

•	All experiments are math benchmarks. It remains unclear how well SeedThink generalizes to open-domain reasoning, multi-hop QA, or commonsense tasks.

•	Although inference is faster, the added step of generating seeds (and running a difficulty estimator) introduces extra pre-processing overhead that’s not quantified.

•	The method assumes fine-grained control at inference time (seed injection, speculative hooks) which may be infeasible in API-only or black-box settings.

•	The claimed “dual-path acceleration” is empirically shown, but theoretical justification (e.g., why token-level alignment should be high) is descriptive, not formal.

**Questions:**

•	How does SeedThink fundamentally differ from Skeleton of Thought beyond adaptive granularity selection? What new capabilities or theoretical insights does it introduce that are not already covered by prior skeleton-based reasoning frameworks?

•	Why were these (*ERCogQA, SpecSearch, and Skeleton of Thought*) baselines excluded from evaluation? Could the authors provide either experimental justification showing how SeedThink compares or complements these approaches?

•	How robust are the reported efficiency gains across different inference engines and deployment configurations? Would the same improvements hold in real-world LLM serving systems with varying compute backends or parallelization schemes?

•	Have you tested SeedThink on non-mathematical reasoning tasks or domains requiring longer context (e.g., legal reasoning, scientific QA)? Does the adaptive seeding still hold there?

•	How robust is the difficulty classifier across tasks or languages? Could a model self-assess difficulty without external classifiers?

•	What is the average compute/time overhead introduced by generating the seed and difficulty estimate? Is total wall-clock time still reduced when including those?

•	How can this method be implemented in commercial APIs or closed models that don’t expose internal “thinking” modes? Would the same approach work with smaller open-weight models?

•	How sensitive are the results to the choice of speculative decoding strategy (e.g., SAM, EAGLE3)? Could speculative gains saturate with longer or more abstract seeds?

•	Are there cases where the seed actually misguides reasoning (e.g., strong but wrong priors)? How does SeedThink recover from such errors?

---

> ### Author Response · Authors · 2025-11-28
> **Response to Reviewer 8k81 [1/9]**
>
> We sincerely thank the reviewer for the time and effort spent reviewing our paper and for acknowledging the novelty of our adaptive seeding approach, the strong empirical results, and the broader potential of our framework. Below, we address the raised questions and concerns in detail:
>
> > **W1:** The proposed “outline” seeding policy in SeedThink appears conceptually similar to the Skeleton of Thought technique presented at ICLR 2024. Although the authors briefly acknowledge this, the overlap in motivation and mechanism warrants deeper discussion.
> > **Q1:** How does SeedThink fundamentally differ from Skeleton of Thought beyond adaptive granularity selection? What new capabilities or theoretical insights does it introduce that are not already covered by prior skeleton-based reasoning frameworks?
>
> **A1:** We thank the reviewer for the opportunity to clarify this distinction. SeedThink and Skeleton of Thought (SoT) differ fundamentally in their goals and mechanisms.
>
> SoT primarily achieves acceleration via parallelism, a strategy that **relies on decomposing a task's final output into independent segments** that can be generated concurrently. Its main benefit is a reduction in wall-clock time through parallel processing, which may not necessarily result in a decrease in the total number of tokens generated.
>
> In contrast, SeedThink is designed to improve the efficiency of the reasoning process itself. Our "seed" is not a structural outline for parallel work, but a reasoning anchor that guides and constrains a **single, sequential** thought process. While SoT focuses on parallelizing the generation of the final output, SeedThink aims to make the reasoning path more direct and concise, a distinct contribution that is particularly effective for tasks with complex, interdependent steps.
>
> > **W2:** The study omits direct comparisons with several key test-time reasoning or preemptive inference strategies such as ERCogQA, SpecSearch, and Skeleton of Thought. These represent state-of-the-art techniques that also focus on optimizing inference efficiency and reasoning control.
> > **Q2:** Why were these (ERCogQA, SpecSearch, and Skeleton of Thought) baselines excluded from evaluation? Could the authors provide either experimental justification showing how SeedThink compares or complements these approaches?
>
> **A2:** First, we apologize for any confusion regarding "**ERCogQA**." Despite a thorough search, we were unable to locate a paper with this specific name and would be grateful if the reviewer could provide a reference so we can ensure a proper comparison.
>
> Second, regarding **SpecSearch**, while it also uses speculative decoding, its focus and mechanism are different from our work. SpecSearch is a novel model-based speculative decoding method that uses smaller draft models to accelerate the token generation speed within a **tree-search** framework, but it does not shorten the length of the reasoning path. In contrast, SeedThink introduces a "dual-path" acceleration. Moreover, we creatively apply existing model-free, suffix-match-based speculative decoding techniques by repurposing the "seed" itself as a speculative draft; this reuse of the seed is a unique aspect of our framework. While our experiments show this can be synergistically combined with a training-based drafter (eagle3 [1]) to **dynamically switch** between two sources of draft tokens for maximum speedup, it is not a mandatory component of our framework.
>
> Third, our work is also different from **Skeleton of Thought (SoT)**, as discussed in the response to **[W1&Q1]**. The fundamental distinction is that SoT's strategy of parallel generation to reduce latency **is contingent on the task being decomposable**, while SeedThink is designed for reasoning tasks with complex, interdependent steps.
>
> [1] Yuhui Li, Fangyun Wei, Chao Zhang, and Hongyang Zhang. Eagle-3: Scaling up inference acceleration of large language models via training-time test. arXiv preprint arXiv:2503.01840, 2025b.

---

> ### Author Response · Authors · 2025-11-28
> **Response to Reviewer 8k81 [2/9]**
>
> > **W3:** The practical efficiency of SeedThink may depend on the underlying LLM serving environment. For instance, differences in batch scheduling, token parallelism, or speculative decoding infrastructure could significantly affect latency and throughput.
> > **Q3:** How robust are the reported efficiency gains across different inference engines and deployment configurations? Would the same improvements hold in real-world LLM serving systems with varying compute backends or parallelization schemes?
>
> **A3:** We believe our framework's efficiency gains are robust across various deployment configurations, as the main performance benefits of SeedThink are two-fold and are independent from the serving environment.
>
> 1. **Efficiency from reduced generation length (platform-agnostic)**: One of the important sources of our speedup comes from our structured, two-stage reasoning pipeline. By first generating a "reasoning anchor" (the seed), our framework guides the LLM to produce a more direct and less verbose reasoning path. This significantly reduces the total number of tokens that need to be generated (up to 68% in our tests). This is a reduction in the fundamental work required to reach a solution. This benefit is inherent to our framework's design and is realised in any serving system, regardless of its load (low or high QPS) or parallelisation scheme, as generating fewer tokens is universally more efficient.
> 2. **Low-overhead acceleration for compute-bound environments**: We agree with the reviewer that the practical efficiency of speculative decoding is nuanced, as the overhead from a secondary draft model can negate gains in compute-bound systems [1]. Our work **sidesteps** these issues by enhancing suffix matching based speculative decoding (existing work like SuffixDecoding [2], SAM decoding [3]). We repurpose the generated "seed"  as a  **higher-quality source for predicting and drafting future tokens** , which improves draft quality and boosts the Mean Acceptance Tokens (MAT) without modifying the verification logic. This approach is inherently low-overhead and training-free. As a result, it effectively mitigates the typical decline in speculative decoding performance seen at larger batch sizes, making it a more viable acceleration strategy for high-QPS environments.
>
> [1] https://blog.vllm.ai/2024/10/17/spec-decode.html
>
> [2] Gabriele Oliaro, Zhihao Jia, Daniel F Campos, and Aurick Qiao. Suffixdecoding: Extreme speculative decoding for emerging ai applications. In The Thirty-ninth Annual Conference on Neural Information Processing Systems, 2025.
>
> [3] Yuxuan Hu, Ke Wang, Xiaokang Zhang, Fanjin Zhang, Cuiping Li, Hong Chen, and Jing Zhang.Sam decoding: Speculative decoding via suffix automaton. arXiv preprint arXiv:2411.10666,2024.

---

> ### Author Response · Authors · 2025-11-28
> **Response to Reviewer 8k81 [3/9]**
>
> > **W4:** The main novelty lies in combining existing paradigms (difficulty estimation, seeding, speculative decoding), not necessarily in new architectures or algorithms.
>
> **A4:** While SeedThink integrates difficulty prediction, seeding, and speculative decoding, we respectfully argue that our contribution is not a simple combination but a **synergistic framework** grounded in a novel empirical insight.
>
> 1. **First, we identified a previously unreported, non-linear relationship between problem difficulty and optimal seed granularity.** Our experiments show that easy problems benefit from detailed seeds that rapidly collapse the search space, whereas hard problems require higher-level seeds to avoid the pitfalls of premature commitment. This empirical discovery is a core contribution and provides the motivation for an **adaptive, difficulty-aware seeding policy** rather than a static one.
>
> 2. **Second, this adaptive seeding provides a new mechanism for optimizing the reasoning process itself.** The seed acts as a compact reasoning anchor that guides the model's exploration, suppresses redundant verification steps, and ultimately shortens the reasoning trajectory. This directly addresses inefficiencies *within* a reasoning process.
>
> 3. **Third, the two-stage structure (Seed -> Think) creates a uniquely effective setup and self-guiding corpus warmup for speculative decoding.** The seed serves as a high-quality, semantically aligned speculative corpus for the subsequent reasoning. This creative reuse of the seed provides a complementary acceleration path without any modification to the underlying speculative decoding algorithm. **The proposed training-free warmup for speculative decoding using generated tokens in the fast thinking mode is novel and not studied in the previous literature.**
>
> In summary, SeedThink is more than the sum of its parts. It uncovers a new empirical behavior in LRM reasoning, operationalizes this insight into an adaptive seeding policy, and leverages the resulting structure to create a dual-path acceleration that is both novel and highly effective.

---

> ### Author Response · Authors · 2025-11-28
> **Response to Reviewer 8k81 [4/9]**
>
> > **W5:** The difficulty estimator is a straightforward text classifier (RoBERTa), not jointly trained or self-adaptive
> > **Q5:** How robust is the difficulty classifier across tasks or languages? Could a model self-assess difficulty without external classifiers?
>
> **A5**:
> **1. On using an external classifier:**
> This was a deliberate design choice motivated by our primary goal: **reducing test-time compute**. Using a lightweight classifier adds only milliseconds of overhead, whereas requiring an LLM to self-assess its confidence or performing joint training would introduce substantial computational costs, undermining our core efficiency objective. Importantly, our framework is modular; the RoBERTa classifier is merely one possible implementation. Any difficulty-estimation method (including more advanced self-assessment models) can be readily integrated without altering the SeedThink framework itself.
>
> **2. On robustness and generalization:**
> We acknowledge this as a limitation: our RoBERTa classifier was fine-tuned specifically for mathematical reasoning, and its performance would likely not generalize well to other domains. However, using a fine-tuned small model for semantic routing is a mature and effective solution in production environments. A prominent example is the vllm-semantic-router [1], a popular open-source library that also relies on fine-tuning a small BERT-style model to route queries. This demonstrates that our approach is a practical and widely used approach. Moreover, adapting seed granularity based on estimated difficulty is task-agnostic and broadly applicable. In practice, training a small, domain-specific classifier is computationally inexpensive, and our framework's logic does not rely on any math-specific assumptions.
>
> **3. On LLM self-assessment:**
> We appreciate this suggestion. While replacing the classifier with LLM self-assessment is a good idea, current models still struggle to achieve reliable, calibrated self-evaluation without costly supervised fine-tuning or RLHF (e.g., [2]). Given these limitations and the significant computational overhead required to enable robust metacognition, we opted for the small external classifier, which provides stable, low-cost, and effective difficulty estimates for our framework.
>
> [1] https://github.com/vllm-project/semantic-router
>
> [2] Yongjiang Liu, Haoxi Li, Xiaosong Ma, Jie Zhang, and Song Guo. Think how to think: Mitigating overthinking with autonomous difficulty cognition in large reasoning models. arXiv preprint arXiv:2507.02663, 2025.

---

> ### Author Response · Authors · 2025-11-28
> **Response to Reviewer 8k81 [5/9]**
>
> > **W6:** All experiments are math benchmarks. It remains unclear how well SeedThink generalizes to open-domain reasoning, multi-hop QA, or commonsense tasks.
> > **Q4:** Have you tested SeedThink on non-mathematical reasoning tasks or domains requiring longer context (e.g., legal reasoning, scientific QA)? Does the adaptive seeding still hold there?
>
>
>
> **A6:** We have conducted new experiments use Qwen3-32B on two challenging, non-mathematical reasoning datasets: **GPQA-Diamond** (complex scientific questions for graduate-level experts) and **FinEval** (expert-level benchmark for finance domain knowledge). These datasets were chosen specifically to test SeedThink's effectiveness in diverse, expert-level domains. Our key findings are listed below:
>
> - **Generalized to new domains**: SeedThink consistently outperforms NoThink and CoThink and achieves the closest accuracy to the Think mode, while persistently delivering a **30-35% token reduction** on both the finance and science datasets. This demonstrates that the core benefit of our framework—maintaining high performance with significantly improved efficiency—is generalizable beyond mathematics.
> - **Regarding adaptive seeding**: Since these domains' datasets do not provide the difficulty labels required to train or validate our difficulty estimator, we used a fixed seed prompt (detailed_steps) for all problems.  We argue that even in this case, the performance of SeedThink remains strong.
>
> **Table 1. Performance on professional finance domain reasoning (FinEval).**
>
> |    Methods    | Average Tokens | Average Runtimes(s) | Accuracy |
> | :-----------: | -------------: | ------------------: | -------: |
> |     Think     |          2,606 |               157.0 |   68.64% |
> |    NoThink    |            498 |                29.5 |   64.29% |
> |    CoThink    |          1,926 |               110.7 |   56.30% |
> | **SeedThink** |          1,814 |                59.9 |   67.94% |
>
> **Table 2. Performance on complex scientific reasoning (GPQA-Diamond).**
>
> |    Methods    | Average Tokens | Average Runtimes(s) | Accuracy |
> | :-----------: | -------------: | ------------------: | -------: |
> |     Think     |          7,643 |               459.3 |   66.16% |
> |    NoThink    |          1,123 |                64.1 |   54.55% |
> |    CoThink    |          5,497 |               315.9 |   58.08% |
> | **SeedThink** |          5,013 |               137.6 |   62.63% |

---

> ### Author Response · Authors · 2025-11-28
> **Response to Reviewer 8k81 [6/9]**
>
> > **W7:** Although inference is faster, the added step of generating seeds (and running a difficulty estimator) introduces extra pre-processing overhead that’s not quantified.
> > **Q6:** What is the average compute/time overhead introduced by generating the seed and difficulty estimate? Is total wall-clock time still reduced when including those?
>
> **A7:** We would like to clarify that all reported acceleration and latency metrics in our paper already account for the full end-to-end process, including the overhead from both the difficulty estimator and the seed generation step. Reported tokens in main table including seed length. Our reported speedups are genuine wall-clock time reductions compared to the baselines.
>
> To provide the explicit quantification the reviewer requested, we have broken down the overhead as follows:
> 1. Difficulty Estimator Overhead: This cost is fixed and minimal. Our lightweight **RoBERTa-based classifier** adds a consistent overhead of only **1-2 milliseconds per query** on our single A100 80G GPU machine. This is negligible compared to the overall latency of LRM generation.
> 2. Seed Generation Overhead: The reviewer correctly intuits that this overhead is more dynamic. As a core feature of our adaptive framework, the cost of seed generation is intentionally variable.  We provide a  statistical analysis of the seed length (and thus its overhead) as it varies with problem difficulty and seed generation prompts in the following table.
>
> **Table 3. Different seed prompts and their generated seed lengths of Qwen3-14B on Math500 dataset.**
>
> |         Seeds         | level-1 | level-2 | level-3 | level-4 | level-5 |
> | :-------------------: | :-----: | :-----: | :-----: | :-----: | :-----: |
> |     outline seed      |   137   |   150   |   179   |   203   |   243   |
> |    key_steps seed     |   274   |   333   |   415   |   514   |   711   |
> | general_approach seed |   302   |   394   |   521   |   912   |  1,391  |
> |  detailed_steps seed  |   585   |   685   |   862   |  1,110  |  1,461  |

---

> ### Author Response · Authors · 2025-11-28
> **Response to Reviewer 8k81 [7/9]**
>
> > **W8:** The method assumes fine-grained control at inference time (seed injection, speculative hooks), which may be infeasible in API-only or black-box settings.
> > **Q7:** How can this method be implemented in commercial APIs or closed models that don’t expose internal “thinking” modes? Would the same approach work with smaller open-weight models?
>
> **A8:** We thank the reviewer for raising these questions about the practical applicability of SeedThink in different deployment scenarios.
>
> **1. Feasibility in black-box / API-only settings:**
>
>  We agree that speculative decoding (SD) is not directly accessible in current commercial API settings. However, **the core SeedThink framework** does not rely on internal model states and remains **fully compatible with API-only or black-box deployments**. Even models that don’t expose internal “thinking” modes, we can seed the first API call for an instruct model to efficiently generate the seed. We note that the inability to run SD is a **general constraint of all black-box APIs**. In contrast, our target scenario is **local or provider-side LLM deployment**, where SD is standard and fully accessible.
>
> **2. Applicability to smaller open-weight models:**
>
> To validate this, we have conducted experiments on Qwen3-1.7B. The results listed in the table below show that even at this smaller scale, SeedThink achieves a **22-24% token reduction** and up to **× 3.37 speed up** when using SAM decoding and eagle3.
>
> **Table 4. Qwen3-1.7B on Math500 dataset, SD short for speculative decoding.**
>
> |       Method/<br/>Length        | Average Runtime (s) |  speed up  |     Average Tokens      | Accuracy |
> | :-----------------------------: | :-----------------: | :--------: | :---------------------: | :------: |
> |              Think              |        147.6        |     -      |          4,846          |  90.00%  |
> |             NoThink             |        25.1         |     -      |           837           |  73.00%  |
> |             CoThink             |        115.1        |   × 1.28   |   268 + 3,980 = 4,248   |  88.20%  |
> |     **SeedThink (w/o SD)**      |        102.7        |   × 1.44   |   553 + 3,234 = 3,787   |  87.80%  |
> |     **SeedThink (w/ SAM)**      |        66.1         |   × 2.23   |   534 + 3,370 = 3,904   |  87.60%  |
> | **SeedThink (w/ SAM + eagle3)** |      **43.8**       | **× 3.37** | 582 + 3,129 = **3,711** |  87.80%  |

---

> ### Author Response · Authors · 2025-11-28
> **Response to Reviewer 8k81 [8/9]**
>
> > **W9:** The claimed “dual-path acceleration” is empirically shown, but theoretical justification (e.g., why token-level alignment should be high) is descriptive, not formal.
>
> **A9:** In the revised manuscript, we will present a more rigorous argument in Section 4.4, showing that high alignment is not an empirical coincidence, but rather a structural consequence of SeedThink’s design.
>
> > **Q8:** How sensitive are the results to the choice of speculative decoding strategy (e.g., SAM, EAGLE3)? Could speculative gains saturate with longer or more abstract seeds?
>
> **A10:** We clarify that SeedThink improves speculative decoding performance through high token-level alignment produced by reusing the seed as the speculative corpus, which notably **strengthens model-free, suffix-match-based speculative methods,** such as SuffixDecoding and SAM decoding, which benefit directly from increased suffix match source. **SeedThink provides substantial acceleration without introducing noticeable overhead.** At the same time, SeedThink is fully compatible with model-based or training-based speculative methods (like Eagle3). While these methods introduce additional drafter models or layers, SeedThink can **dynamically switch** between two sources of draft tokens, thus achieving a higher end-to-end speedup.
>
> Regarding the concern about potential saturation with longer or more abstract seeds, our new experiment show in the table below indicate the opposite trend: **longer seeds generally increase speculative gains** (speed up from **1.47 to 1.64** and MAT from **1.65 to 1.96**), as they produce more opportunities for suffix matches. **We do not observe saturation within the studied range**, and we note that seed length is intentionally constrained: excessively long seeds would undermine the two-stage pipeline’s efficiency. Within this realistic range, speculative gains(Mean accept tokens) grow monotonically.
>
> **Table 5.  Speculative gains (using SAM decoding) of different seed prompt Qwen3-14B on the Math500 dataset，MAT short for Mean Accept Token**
>
> |               Seeds               | MAT  | Throughput(tokens/s) | Speed up from SD | Length | Accuracy |
> | :-------------------------------: | :--: | :------------------: | :--------------: | :----: | :------: |
> |          Think (w/o SAM)          |  0   |        28.44         |        -         | 4,741  |  96.8%   |
> |          Think (w/ SAM)           | 1.65 |        40.26         |      × 1.42      | 4,776  |  96.8%   |
> |     SeedThink (outline seed)      | 1.73 |        41.72         |      × 1.47      | 2,959  |  93.8%   |
> |    SeedThink (key_steps seed)     | 1.82 |        43.88         |      × 1.54      | 2,847  |  93.4%   |
> | SeedThink (general_approach seed) | 1.88 |        45.04         |      × 1.58      | 2,735  |  95.6%   |
> |  SeedThink (detailed_steps seed)  | 1.96 |        46.87         |      × 1.64      | 2,650  |  94.4%   |

---

> ### Author Response · Authors · 2025-11-28
> **Response to Reviewer 8k81 [9/9]**
>
> > **Q9:** Are there cases where the seed actually misguides reasoning (e.g., strong but wrong priors)? How does SeedThink recover from such errors?
>
> **A11:** Yes, that can happen. A seed with a strong but flawed premise can sometimes misguide the model's reasoning process.
>
> In many cases, SeedThink can recover from such errors for two key reasons:
>
> - The model possesses an inherent ability to identify biased information.
>
> - Our reasoning stage prompt is specifically designed to compel the model to verify the seed's contents and make necessary corrections if it finds any bias.
>
> For a more quantitative analysis of this situation, we conducted an experiment to generate 4*500 seeds by our SeedThink method using various seed prompts on the Math500 dataset. We categorized each seed based on its content and its potential to introduce bias:
>
> - **Already correct:** The seed itself contains the final correct answer, making the subsequent reasoning a process of formal verification.
> - **No biased information:** The seed provides a direct path toward the correct answer, without containing misleading or incorrect details. A more detailed seed offers more relevant information and analytical steps.
> - **Introduce biased information but not affect:** The seed contains errors, but the LRM, during its reasoning phase, identifies and corrects this flaw, arriving at the correct final answer. This demonstrates the model recover from such errors.
> - **Introduce biased information and affect:** The seed contains errors that mislead the LRM, causing the final answer to be incorrect. This represents the  scenario the reviewer is concerned about.
>
> **Table 6. Analysis of various seed prompts and their effect on final answers.**
>
> |    **Seed Prompt**    | **Already Correct** | **No Biased Info** | **Biased Info, No Affect** | **Biased Info, Affects Final Answer** |
> | :-------------------: | :-----------------: | :----------------: | :------------------------: | :-----------------------------------: |
> |     outline seed      |          0          |   100% (500/500)   |             0              |                   0                   |
> |    key_steps seed     |   11.2% (56/500)    |  75.2% (376/500)   |       8.0% (40/500)        |             5.6% (28/500)             |
> | general_approach seed |   50.8% (254/500)   |  31.2% (156/500)   |       12.8% (64/500)       |             5.2% (26/500)             |
> |  detailed_steps seed  |   81.2%(406/500)    |   3.4% (17/500)    |       8.4% (42/500)        |             7.0% (35/500)             |
>
> This analysis leads to two key observations:
>
> 1. **Cases of misguidance are infrequent.** The rightmost column shows that the number of instances where a flawed seed leads to an incorrect final answer is low, peaking at 7.0% for the most detailed seed types. Furthermore, we should also clarify that the above result comes from using a single seed type across all problem difficulties. In practice, our SeedThink strategy is adaptive: we only use detailed_step seeds for simple problems, where they are less likely to mislead the model.
> 2. **The results demonstrate that SeedThink possesses a self-correction capability**. The third column ("Biased Info, No Affect") shows that the model is not a naive follower. For example, when presented with 90 flawed seeds from the general_approach , it still adjusted its reasoning path to produce a correct final answer in 64 of those cases (**a 71% success rate**).

---

### Author Response · Authors · 2025-12-04
**Letter to all reviewers and area chairs**

We would like to **express our deep gratitude to all reviewers and Area Chairs** for their time and effort in reviewing our manuscript. We thoroughly valued all feedback and conducted significant clarification with new analysis to address the reviewers' comments in our response.

**Recapping major contributions of our work,** we first uncover a novel, non-linear relationship between problem difficulty and optimal seed granularity, a previously unreported empirical insight in LRM reasoning. With this discovery, we systematically reveal the limitations of static seeding policies and motivate the need for an adaptive, difficulty-aware approach. Based on this, we propose a general framework, SeedThink, which consists of an adaptive seeding policy and a two-stage reasoning structure to realize a dual-path acceleration. We present extensive and comprehensive experimental results to demonstrate its effectiveness in optimizing the reasoning process itself and in creating a uniquely effective setup for speculative decoding.

Given the reviewing feedback, we are **encouraged to receive numerous positive recognition**, including:
- The core insight of adapting seed granularity to problem difficulty is novel and powerful. Reviewers highlighted that the systematic demonstration of the non-linear relationship between difficulty and optimal seed granularity is a valuable contribution that moves beyond one-size-fits-all approaches (Reviewers 2XUz, mFyz, 8k81).
- The proposed SeedThink is a clear and effective framework for accelerating reasoning. The two-stage structure was praised for its novelty and ability to shorten reasoning trajectories, with reviewers commending the creative reuse of the seed as a high-quality draft for speculative decoding, enabling a dual-path acceleration (Reviewers 7GgU, 8k81, 2XUz).
- The experiments are comprehensive and convincing, demonstrating significant efficiency gains with minimal accuracy degradation. Reviewers acknowledged the strong empirical results, including up to 4.1x end-to-end acceleration and a 68% reduction in token length on multiple math benchmarks, showcasing a superior efficiency-accuracy trade-off (Reviewers 8k81, mFyz, 7GgU).

More importantly, given the reviewers' valuable comments and suggestions, **our manuscript has been continually improved**. We carefully followed the reviewers' comments to revise the manuscript with the following:
- Additional experiments on non-mathematical domains (GPQA-diamond, FinEval) and more models to demonstrate the framework's generalizability and applicability (Reviewers 8k81, 2XUz, mFyz).
- Additional experiments and analysis on system-level throughput in multi-batch serving scenarios to validate practical efficiency gains beyond single-instance latency (Reviewer mFyz).
- Deeper discussion and comparison with related work such as Skeleton of Thought and SpecSearch, clarifying the novelty of our approach (Reviewer 8k81).
- A more rigorous justification for the high token-level alignment that enables dual-path acceleration, showing it's a structural consequence of our design (Reviewer 8k81).
- Quantitative analysis of the difficulty classifier's performance and the impact of misclassifications, demonstrating the pipeline's robustness (Reviewers mFyz, 7GgU).
- Quantitative analysis of how the model recovers from incorrect seeds, highlighting its self-correction capabilities (Reviewers 7GgU, 8k81).
- Ablation study on the impact of seed length on speculative decoding gains, clarifying the mechanism and showing that gains do not saturate (Reviewer 8k81).
- Explicit quantification of pre-processing overhead from the difficulty estimator and seed generation, confirming that reported speedups are end-to-end (Reviewer 8k81).
- Clarification of the speculative decoding implementation and its role in accelerating the later reasoning stage (Reviewers mFyz, 7GgU).
- Revise the visualization, dense presentation, and other minor typos (Reviewers 7GgU).

**Again, we deeply value the constructive critique and advice the reviewers provided, as well as the attention and effort of the area chairs.** We respectfully hope that our rebuttal efforts can be taken into proper account.

Best regards,

Authors of #25520

---

### Meta-Review · Area_Chair_NYLn · 2026-01-02

**Summary:**

This paper proposes SeedThink, a test-time framework for improving reasoning efficiency in large reasoning models via adaptive seed-thought initialization. The key insight is that the optimal granularity of the initial seed depends on problem difficulty, and that such seeds can both guide reasoning toward shorter trajectories and be reused to enhance speculative decoding, enabling dual-path acceleration.

The paper received four reviews with mixed scores, including one reject (mFyz), two borderline below threshold (8k81, 7GgU), and one borderline above threshold (2XUz). While reviewers agree that efficient reasoning is an important problem and that the empirical results are promising, several core concerns remain unresolved after the rebuttal, particularly regarding novelty, robustness and generalization of the multi-stage pipeline.

**Reviewer Concerns:**

Reviewer 8k81 raised concerns about novelty relative to prior work such as Skeleton of Thought, as well as the lack of theoretical justification for the proposed dual-path acceleration. These concerns are mainly addressed.

Reviewer 2XUz questioned the generalization of the method beyond mathematical reasoning. The authors added experiments on finance and scientific reasoning benchmarks. These concerns are mainly addressed.

Reviewer mFyz expressed strong concerns about the fragility of the multi-stage pipeline, reliance on an external classifier, system-level overhead, and prompt sensitivity. While the rebuttal provided additional analyses and throughput experiments, the method remains dependent on multiple hand-designed components, and its robustness under realistic deployment settings is still uncertain.

Reviewer 7GgU raised issues regarding clarity and speculative decoding integration. These concerns were addressed through improved explanations and additional analysis.

**Reviewer Scores:**

The paper received one reject (mFyz), two borderline below threshold (8k81, 7GgU), and one borderline above threshold (2XUz).

Reviewer 7GgU would likely increase his score to a borderline-accept score, as their concerns were largely addressed.

Reviewer 2XUz would likely maintain a borderline-accept score, since added experiments partially address generalization concerns.

Reviewer 8k81 would likely increase his score to a borderline-accept score, as concerns about novelty and theoretical grounding are largely resolved.

Reviewer mFyz would likely maintain a borderline-reject score, given concerns about pipeline robustness and practicality.

---

### Decision · Program_Chairs · 2026-01-26

Reject